# The High Osmolarity Glycerol Mitogen-Activated Protein Kinase regulates glucose catabolite repression in filamentous fungi

Leandro José de Assis[1], Lilian Pereira Silva[1], Li Liu[2], Kerstin Schmitt[2], Oliver Valerius[2], Gerhard H. Braus[2‡]*, Laure Nicolas Annick Ries[3‡]*, Gustavo Henrique Goldman[1,4‡]*

1 Faculdade de Ciências Farmacêuticas de Ribeirão Preto, Bloco Q, Universidade de São Paulo, Brazil, 2 Department of Molecular Microbiology and Genetics and Goettingen Center for Molecular Biosciences (GZMB), University of Goettingen, Goettingen, Germany, 3 Faculdade de Medicina de Ribeirão Preto, Universidade de São Paulo, Brazil, 4 Institute for Advanced Study, Technical University of Munich, Garching, Germany

‡ GHB, LNAR, and GHG are joint senior authors on this work.
* gbraus@gwdg.de (GHB); rieslaure13@gmail.com (LNAR); ggoldman@usp.br (GHG)

**Data Availability Statement:** All relevant data are within the manuscript and its Supporting Information files.

## Abstract

The utilization of different carbon sources in filamentous fungi underlies a complex regulatory network governed by signaling events of different protein kinase pathways, including the high osmolarity glycerol (HOG) and protein kinase A (PKA) pathways. This work unraveled cross-talk events between these pathways in governing the utilization of preferred (glucose) and non-preferred (xylan, xylose) carbon sources in the reference fungus *Aspergillus nidulans*. An initial screening of a library of 103 non-essential protein kinase (NPK) deletion strains identified several mitogen-activated protein kinases (MAPKs) to be important for carbon catabolite repression (CCR). We selected the MAPKs Ste7, MpkB, and PbsA for further characterization and show that they are pivotal for HOG pathway activation, PKA activity, CCR via regulation of CreA cellular localization and protein accumulation, as well as for hydrolytic enzyme secretion. Protein-protein interaction studies show that Ste7, MpkB, and PbsA are part of the same protein complex that regulates CreA cellular localization in the presence of xylan and that this complex dissociates upon the addition of glucose, thus allowing CCR to proceed. Glycogen synthase kinase (GSK) A was also identified as part of this protein complex and shown to potentially phosphorylate two serine residues of the HOG MAPKK PbsA. This work shows that carbon source utilization is subject to cross-talk regulation by protein kinases of different signaling pathways. Furthermore, this study provides a model where the correct integration of PKA, HOG, and GSK signaling events are required for the utilization of different carbon sources.

## Author summary

Filamentous fungi secrete an array of biotechnologically valuable enzymes, with enzyme production being inhibited in the presence of preferred carbon sources, such as glucose,

**Funding:** This work was supported by the Fundação de Amparo à Pesquisa do Estado de São Paulo (FAPESP) 2014/00789-6 and 2017/23624-0 to LJA; FAPESP 2016/07870-9 and Technical University of Munich-Institute for Advanced Study (TUMIAS) to GHG; German Research Council (DFG grant BR1502/19-1) to LL and GHB. Funding for Open Access publication supported by Göttingen University. The funders had no role in study design, data collection and analysis, decision to publish, or preparation of the manuscript.

**Competing interests:** The authors have declared that no competing interests exist.

in a process known as carbon catabolite repression (CCR). This work unravels upstream signalling events that regulate CCR in *Aspergillus nidulans*. Different mitogen-activated protein kinases (MAPKs) were identified and shown to be crucial for CCR and protein kinase A (PKA) activity, which is essential for carbon source utilisation in filamentous fungi. Furthermore, the MAPKs formed a protein complex with additional protein kinases, such as glycogen synthase kinase (GSK), which is important for glucose metabolism; resulting in the inhibition of CCR in the presence of non-preferred carbon sources. GSK was shown to potentially phosphorylate the MAPK PbsA of the high osmolarity glycerol (HOG) pathway. This study thus unravels the cross-talk between protein kinases from different signalling pathways that regulate carbon source utilisation in filamentous fungi.

## Introduction

Protein phosphorylation, which is catalysed by protein kinases, is crucial for target protein function and/or cellular localization, resulting in the regulation of a variety of cellular processes and signalling pathways. In the filamentous fungus *A. nidulans*, a role for protein kinases in the regulation of carbon utilisation pathways, such as carbon catabolite repression (CCR), has been shown [1]. CCR is a mechanism by which fungi use the energetically most favourable carbon source (e.g. glucose), and in *A. nidulans*, this process is regulated by the transcription factor (TF) CreA. Homologues of CreA are present in other filamentous fungi, including *Trichoderma reesei*, *Neurospora crassa*, *A. flavus*, *A. niger*, and *A. fumigatus*, where this TF is also important for the regulation of genes encoding hydrolytic enzymes, such as xylanases and cellulases, as well as for the use of alternative carbon sources and glucose metabolism [2–6]. In the presence of glucose, CreA localizes to the nucleus, where it represses genes required for the utilisation of non-glycolytic carbon sources, whereas the absence of glucose causes translocation of CreA to the cytoplasm [7].

Several studies have supported the role of phosphorylation in the post-translational regulation of CreA [8–11]. In *Saccharomyces cerevisiae* and *A. nidulans*, the AMP-activated protein kinase Snf1p/SnfA regulates cellular localization of Mig1p (CreA homologue)/CreA. Under glucose stress or starvation conditions, Snf1p is activated by phosphorylation at threonine 210, resulting in the subsequent translocation of Snf1p to the nucleus, where it phosphorylates Mig1p [12,13]. Subsequently, Mig1p re-localizes to the cytoplasm, relieving gene repression and allowing induction of genes encoding enzymes required for alternative carbon source utilisation [12–14]. In *A. nidulans*, SnfA regulates the assimilation and utilization of alternative carbon sources [1]. The deletion of *snfA* causes CreA to permanently reside within the nucleus even in carbon catabolite (CC)-de-repressing conditions [1]. Furthermore, the role of the *A. nidulans* cAMP-dependent protein kinase A (PKA) catalytic subunit PkaA in the regulation of CreA, has been shown. Deletion of *pkaA* results in CC-de-repression, even in the presence of glucose, due to aberrant CreA cellular localization and glucose signalling, therefore increasing the amount of secreted hydrolytic enzymes in both CC-repressing and de-repressing conditions [15]. In *T. reesei*, phosphorylation of CRE1 (CreA homologue) at serine 241, is catalyzed by casein kinase and is required for stabilizing CRE1 in the nucleus [10]. In *A. nidulans*, CreA was shown to be phosphorylated directly at serine 262 by casein kinase A (CkiA) and indirectly at serine 319 by PkaA in the presence of glucose, suggesting that CreA phosphorylation is required for repression in this fungus [9,11].

Adding to the complexity of the regulation of carbon source utilisation is the interaction of different cellular signalling pathways. Glucose signalling and CCR are intrinsically connected, with the phosphorylation of glucose during the first step of glycolysis serving as a signal for the activation of CCR [16,17]. In *S. cerevisiae*, the absence of glucose phosphorylation in the triple protein kinase deletion mutant *ΔGLK1 ΔHXK1 ΔHXK2*, results in a reduction of Ras2 activity, cAMP induction and PKA activity [18], since cAMP is required for PKA activation through binding to the PKA regulatory subunit [19]. In *A. nidulans*, the deletion of *pkaA* impairs hexo/ glucokinase activities and glucose transport, further supporting the importance of the PKA pathway for glucose metabolism and CCR [15]. In some plant pathogenic fungi such as *Alternaria brassicicola*, *Cochliobolus heterostrophus*, and *Fusarium oxysporum*, mitogen-activated protein kinase pathways (MAPKs) were shown to play a role in the secretion of hydrolytic enzymes [10,20,21]. In *N. crassa*, the High Osmolarity Glycerol (HOG) MAPK pathway senses the presence of free soluble sugars and regulates the expression of genes required for the use of alternative carbon sources [22]. In *T. reesei*, the HOG MAPK pathway was reported to be involved in the induction of cellulase-encoding genes, with the absence of the MAPKs TMK1 and TMK2 resulting in increased cellulase production even if fungal growth was impaired [23,24].

Cross-talk between signaling pathways, such as the one between the PKA and HOG MAPK pathways, in response to extracellular stresses and stimuli, appears to be conserved from yeast to mammalian cells. In mammalian neuronal cells, cAMP/PKA signaling results in the activation of the MAPK pathway, resulting in the regulation of plasticity-associated genes [25,26]. Furthermore, the presence of glucose activates the insulin pathway, resulting in the phosphorylation and inactivation of glycogen synthase kinase 3 (GSK3), which in turn allows glycogen synthesis, catalyzed by glycogen synthase a, to proceed [27]. In fungi, glycogen and trehalose metabolism are controlled by GSK and the PKA pathway, suggesting a conservation of function for PKA in the regulation of glycogen metabolism [6,28,29].

The aforementioned studies suggest a complex interplay of different protein kinase signaling pathways, that are involved in fungal carbon source utilization, and that coordinately have not extensively been investigated in one single fungal species. This work therefore aimed at elucidating the signaling events that govern the utilization of preferred and alternative carbon sources in the reference filamentous fungus *A. nidulans*. This study provides strong evidence for cross-talk between the HOG and PKA pathways in regulating sugar utilization and shows that GSK is also important for these events. We propose a model whereby the correct integration of signaling events and cascades that constitute the different pathways are crucial for glucose-mediated CCR as well as CC-de-repression and subsequent secretion of hydrolytic enzymes that are required for alternative carbon source utilization in *A. nidulans*.

## Results

### Mitogen-activated protein kinases (MAPK) are important for CCR

In order to determine which protein kinases are important for CCR, an *A. nidulans* library, containing 103 non-essential protein kinase (NPK) deletion strains [30], was screened for growth in the presence of 2-deoxyglucose (2DG) and allyl alcohol (AA). 2DG and AA are indicators for defects in CCR [31,32], with 2DG being a glucose analogue that cannot be metabolized, and AA being converted to the highly toxic compound acrolein by alcohol dehydrogenase (ADH) [33–35]. Growth phenotypes of 2DG- and AA- resistant or sensitive strains were confirmed by radial growth on plates and compared to the wild-type strain in the condition (Fig 1A and 1B). We identified nine protein kinase deletion strains (*ΔatmA, Δoca2, Δkin1, Δste7, ΔmpkB, ΔmpkC, Δppk33, ΔsnfA*, and *Δscy1*) that were sensitive to at least one

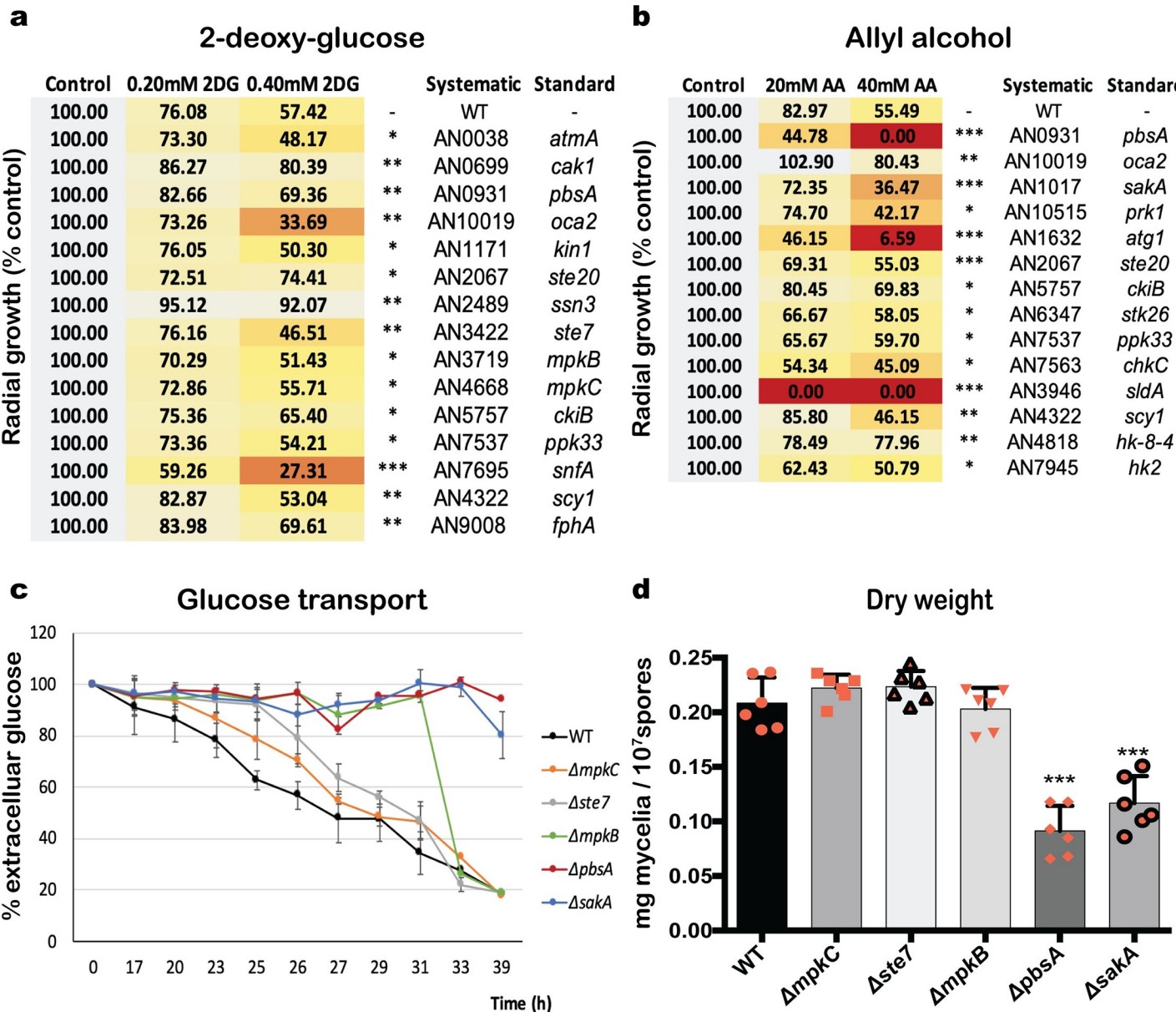

**Fig 1. Mitogen-activated protein kinases (MAPKs) are important for CCR.** Heat map and values depicting average radial growth of three biological replicates of protein kinase deletion strains that were significantly sensitive or resistant to at least one concentration of 2-deoxyglucose (2DG) (a) or allyl alcohol (AA) (b). Strains were grown from $10^5$ spores on glucose or xylose minimal medium (MM) supplemented with increasing concentrations of 2DG and AA for 5 days at 37°C before colony diameter was determined. Growth is given as the percentage in comparison to the control (without 2DG, AA) condition for each strain. Extracellular glucose concentrations were determined in culture supernatants of all MAPK deletion strains grown from $10^7$ spores after transfer from complete medium (24 h) to glucose MM for 39 h at 37°C (c). Also shown is the fungal dry weight of the same strains after 39 h (d). Standard deviations represent the average of six biological replicates, with *p-value<0.05, **p-value<0.001, ***p-value<0.0001 in a two-way ANOVA multiple comparison test.

concentration of 2DG and six strains (Δcak1, ΔpbsA, Δste20, Δssn3, ΔckiB and ΔfphA) that were resistant to 2DG (Fig 1A). Furthermore, eleven protein kinase deletion strains (ΔpbsA, ΔsakA, Δprk1, Δatg1, Δste20, Δstk26, Δppk33, ΔchkC, ΔsldA, Δscy1, and Δhk2) were sensitive to at least one concentration of AA; and three strains (Δoca2, ΔckiB and Δhk-8-4) were resistant to AA (Fig 1B). In the cell, AA is converted to acrolein, a potent inducer of oxidative stress [36–38]. To determine whether the observed strain phenotypes could also be related to defects

in the oxidative stress response, we assessed resistance and sensitivity of these strains to acrolein (S1 Fig). Strains Δ*pbsA*, Δ*sakA*, Δ*prk1*, Δ*atg1*, Δ*ste20*, Δ*ckiB*, Δ*stk26*, Δ*sldA*, Δ*scyA*, and Δ*hk-8-4* were sensitive to increased concentrations of acrolein (S1 Fig). These strains, with the exception of the Δ*ckiB* and Δ*hk-8-4* strains were also sensitive to AA (S1 Fig). These results therefore suggest that the observed strain resistance/sensitivity phenotypes to AA could be due to either defects in CCR and/or oxidative stress responses.

Of particular interest was the presence of several 2DG- and/or AA- sensitive and resistant strains that were deleted for genes encoding protein kinases of different MAPK pathways. To further determine the role of these MAPK pathway deletion strains in CCR, glucose transport was measured in all NPKs, that are part of a MAPK pathway, and sensitive/resistant to 2DG and AA. It has been shown that increased resistance or sensitivity to 2DG can be due to a reduction in glucose uptake in a mutation-dependent manner [39,40] in addition to other 2-DG off-target effects [41]. Furthermore, the first step of glucose phosphorylation is essential for CCR regulation and it is controlled by the PKA pathway, another important regulator of CCR [1,15]. Glucose transport and growth in the presence of glucose was significantly reduced in two deletion strains (Δ*pbsA* and Δ*sakA*), with another three strains (Δ*ste7*, Δ*mpkB*, and Δ*mpkC*) showing a delay in glucose transport but no growth-associated phenotype (Fig 1C and 1D). These results suggest that the significant reduction in glucose transport in the Δ*pbsA* and Δ*sakA* may be related to growth impairments in this carbon source. Nevertheless, several MAPK pathway kinases appear to be important for glucose metabolism in *A. nidulans*.

## Different MAPKs are important for HOG pathway activation and PKA activity

To further confirm the importance of MAPK pathways for CCR, we established a network interaction profile of the 23 protein kinases, whose deletion resulted in a significantly altered growth phenotype in the presence of 2DG and AA, by using String (https://string-db.org/). Thirteen of these 23 proteins showed different degrees of network interactions (direct physical and/or indirect genetic) interactions are presented as arrows in the direction of the target and formed two groups: the first group is composed by *oca2* (AN10019), *atmA* (AN0038), *chkC* (AN7563), *cak1* (AN0699) and *sldA* (AN3946); whereas the second group comprises *ste7* (AN3422), *mpkB* (AN3719), *pbsA* (AN0931), *ste20* (AN2067), *sakA* (AN1017), *mpkC* (AN4668), *snfA* (AN7695) and *ssn3* (AN2489) (Fig 2A). The second group contains the two HOG MAPKs MpkC and SakA, that are potentially interacting with the protein kinase SnfA, which was shown to be a key regulator of CCR [1] (Fig 2A).

Accordingly, we decided to investigate in more detail the involvement of the three MAPK pathway-related kinases Ste7, MpkB and PbsA in the regulation of CCR, in order to determine the regulatory influence of non-terminal, upstream MAPK pathway proteins for the utilization of preferred and alternative carbon sources. The MAPK SakA that belongs to the HOG pathway under the effect of red light and/or osmotic stress is activated by phosphorylation moving to the nucleus and it has as target the transcription factor *atfA* that mediates the upregulation of several genes involved in osmotic stress response, oxidative stresses, development and cell wall [42–45]. In a first instance, HOG pathway activation was determined by using the anti-phospho-p38 antibody to detect phosphorylated SakA by Western blot in total protein extracts from the Δ*ste7*, Δ*mpkB* and Δ*pbsA* strains, when grown in CC-de-repressing (xylose) and CCR (in the presence of glucose) conditions. This commercial antibody specifically recognizes phosphorylated SakA since there are no bands in the *A. nidulans* Δ*sakA* strain (Fig 2B). In the wild-type (WT) strain, SakA is phosphorylated in the presence of xylose and this phosphorylation is highly reduced when glucose is added (Fig 2B). To normalize the observed strain-

**a**

## Kinases network interactions

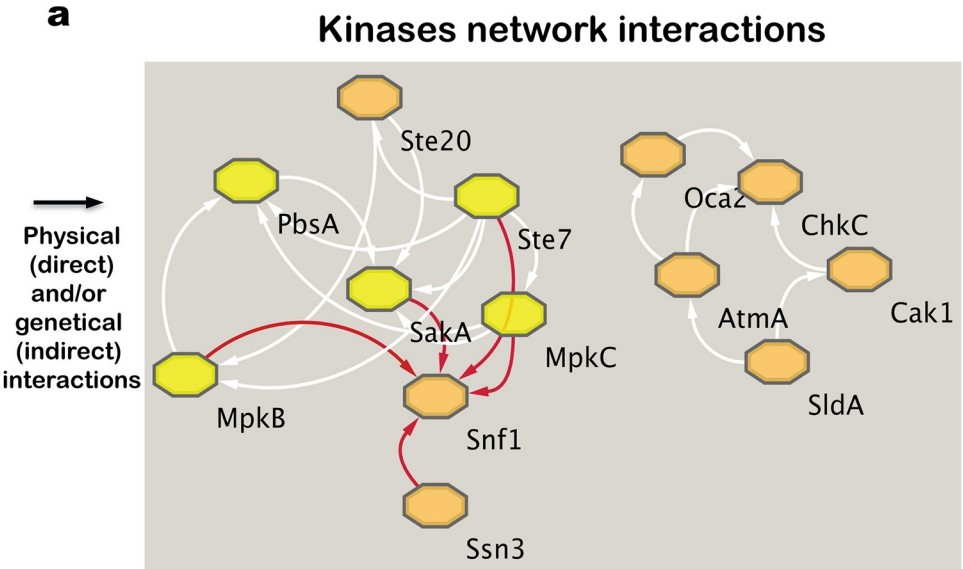

Physical
(direct)
and/or
genetical
(indirect)
interactions

**b**

## SakA phosphorylation

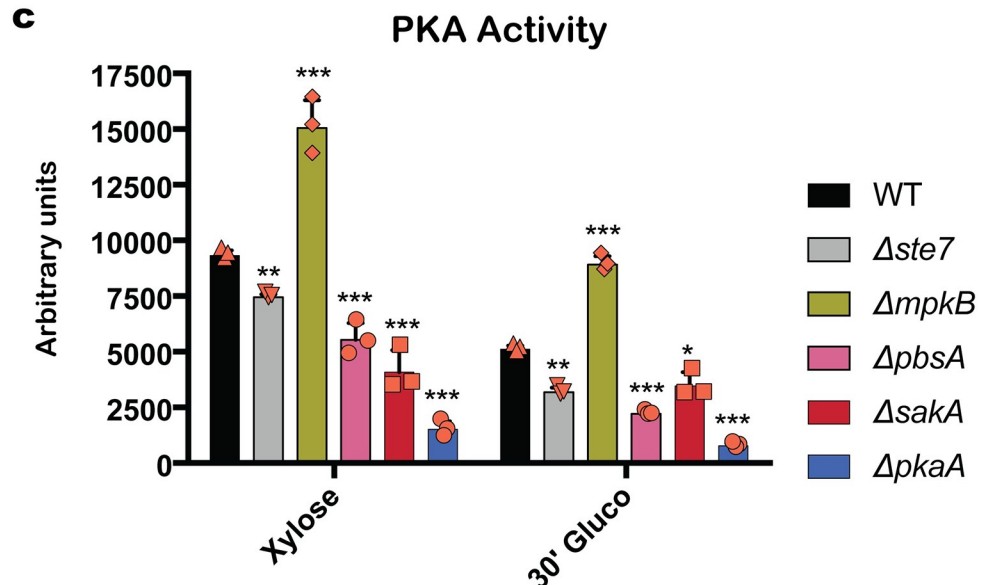

**Fig 2. The protein kinases Ste7, MpkB and PbsA are important for HOG and PKA pathway activation.** STRING network analysis showing direct physical and/or indirect genetic interactions (arrows) of the 27 protein kinases, which were identified as being involved in CCR, results in two separate groups. Protein connections are depicted in a diagram having omitted all proteins with no predicted interaction. Arrows indicate the direction of the interaction that can be physical and/or genetic, with red arrows highlighting proteins that interact with SnfA and yellow octagons depicting the core proteins of the network (a). SakA phosphorylation, as a direct measure of HOG pathway activation, was determined by Western blot in the WT, Δste7, ΔmpkB and ΔpbsA strains by using an anti-P-p38 antibody. Strains were grown in xylose MM for 24 h before glucose was added for 30 min, total cellular proteins were extracted and Western blots were carried out. Cellular extracts from the WT and ΔsakA strains grown in glucose were used as controls. SakA phosphorylation levels were normalized by β-actin protein levels and densitometry ratios of P-p38/β-actin are indicated (b). Ste7, MpkB, PbsA and SakA are important for PKA pathway activation. PKA activity was determined in total protein extracts obtained from the WT, Δste7, ΔmpkB, ΔpbsA, ΔsakA and ΔpkaA strains when grown in the same conditions as specified in (b). PKA activity was normalized by total protein extract concentration (c). Standard deviations represent the average of three biological replicates (depicted in orange), with *p-value<0.05, **p-value<0.001, ***p-value<0.0001 in a two-way ANOVA multiple comparison test.

specific phosphorylated SakA levels, we used anti-ß-actin as the antibody that detects total cellular protein since the anti-p38 antibody to detect total SakA does not function for *A. nidulans* cellular protein extracts. The addition of glucose therefore causes inactivation of the HOG pathway, and these results are consistent with studies in *S. cerevisiae* and mammalian cells [46,47]. In contrast, SakA is phosphorylated in both conditions in the *Δste7* and *ΔmpkB* strains, with the phosphorylation signal being much stronger in the *Δste7* strain than when compared to the *ΔmpkB* strain. In addition, SakA phosphorylation was very low or absent in the *ΔpbsA* strain (Fig 2B). PbsA is the MAPKK and scaffold protein [48] that controls SakA phosphorylation, and the here described results therefore confirm that PbsA also controls SakA phosphorylation in *A. nidulans*. (Fig 2B). In addition, we determined cellular localization of SakA in CC-de-repressing (xylan) and repressing (xylan and glucose) conditions by constructing a strain with a single copy of SakA-GFP integrated at the *sakA* locus and under the control of the native promoter. In the presence of xylan, SakA-GFP was predominantly observed in the nuclei (82.6%); whereas the addition of 2% w/v glucose for 30 min, resulted in the partial re-localization of SakA-GFP to the cytoplasm with SakA-GFP fluorescence observed in 52.9% of nuclei (S2 Fig). These results suggest that SakA cellular localization is carbon source-dependent and important for the control of carbon catabolite de-repression in *A. nidulans*.

In *A. fumigatus*, SakA physically interacts with the catalytic subunit of PKA, regulating PKA activity under osmotic stress [28]. Due to the high conservation of function of the HOG and PKA pathways in different fungal species, we wondered whether SakA phosphorylation is also involved in PKA activation in *A. nidulans*. PKA activity was significantly higher in the *ΔmpkB* strain and reduced in the *ΔpbsA* and *ΔsakA* strains in both CCR and de-repression conditions (Fig 2C). Deletion of *sakA* resulted in significantly reduced PKA activity in both conditions (Fig 2C), suggesting that, like in *A. fumigatus*, SakA is also important for PKA activity in *A. nidulans*, although whether this occurs through direct physical interaction remains to be determined. In contrast, PKA activity was significantly reduced in the *Δste7* strain and increased in the *ΔmpkB* strain. This pattern was also observed for SakA phosphorylation levels in these strains (Fig 2B), suggesting that Ste7 and MpkB control PKA activity through other/additional mechanisms and/or regulatory proteins. These results suggest that the protein kinases Ste7, MpkB and PbsA regulate, either directly and/or indirectly, SakA phosphorylation and PKA activity in *A. nidulans*.

## Different MAPKs are important for CreA cellular localization, protein turnover and enzyme production

To further describe a role of MAPK pathways in CCR in *A. nidulans*, microscopy analysis of CreA-GFP in the wild-type, *Δste7*, *ΔmpkB* and *ΔpbsA* background strains was carried out. All

these strains have single copies of CreA-GFP integrated at the *creA* locus and under the control of the native promoter. Strains were grown in MM supplemented with xylan, a complex polysaccharide shown to cause exclusion of CreA-GFP from the nucleus; and after the addition of glucose, which results in CCR and CreA-GFP localization to the nucleus [7] (Fig 3A). In CC-de-repressing conditions (xylan), 29, 40, 15 and 19% of all counted nuclei of the WT, *Δste7*, *ΔmpkB* and *ΔpbsA* strains respectively, contained CreA-GFP. The addition of glucose caused an accumulation of CreA-GFP in the nuclei of the WT and *ΔmpkB* strains (~100%), whereas CreA-GFP nuclear localization was significantly reduced in the *Δste7* (~69%) and *ΔpbsA* (~35%) strains (Fig 3B). These results indicate that the protein kinases Ste7 and PbsA, but not MpkB, are important for CreA cellular localization under CCR condition.

To further support the microscopy studies and to gain understanding of the CreA-GFP protein dynamics, we carried out Western blots of immunoprecipitated CreA-GFP in the different protein kinase deletion strains. We also included the *ΔpkaA* strain, as the PKA pathway was shown to be important for CCR [15]. Similarly to a previous study [31], full length CreA-GFP protein levels (about 76 kDa, indicated by an arrow in Fig 4A) are reduced in the presence of xylose (CC-de-repression) in the WT strain, whereas the addition of glucose (CCR) caused an increase in CreA-GFP protein levels (Fig 4A). CreA-GFP accumulation followed a similar pattern as the WT strain in the *Δste7* strain with a faint amount of protein being present in CC-de-repressing conditions (Fig 4A). In contrast, the *ΔmpkB* strain showed high levels of CreA-GFP protein in all conditions whereas no CreA-GFP was detected in the *ΔpbsA* and *ΔpkaA* strains (Fig 4A). These results suggest that the HOG MAPKs MpkB and PbsA as well as PKA are pivotal for the regulation of CreA-GFP protein turnover (Fig 4A). In summary, protein localization and dynamics studies suggest that Ste7 is important for CreA-GFP localization in CC-repressing conditions but not for CreA-GFP protein synthesis; MpkB is not required for CreA-GFP cellular localization under repressing conditions but crucial for CreA-GFP degradation in CC-de-repressing conditions; PbsA is pivotal for CreA-GFP cellular localization and synthesis in all conditions and PKA is also crucial for CreA-GFP synthesis in both conditions.

In *T. reesei* and *N. crassa*, the absence of the respective SakA homologues resulted in an increase in cellulase and xylanase production [22–24,49], suggesting that the HOG pathway is not only involved in CCR but also in the regulation of hydrolytic enzyme production. We therefore determined xylanase activity in the supernatants of the *A. nidulans* protein kinase deletion strains in both CC-de-repressing (xylose) conditions and CC-repressing (glucose and xylose). In the presence of CCR conditions, xylanase activity was significantly reduced in the *Δste7* and *ΔmpkB* strains and highly induced in the *ΔpbsA* strain, especially after 48 h and 72 h (Fig 4B). Similarly, xylanase activity was significantly reduced in the *ΔmpkB* strain and *Δste7* strain in the time point 72 h and induced in the *ΔpbsA* strain in the presence of xylose, whereas enzyme activity was not significantly different for the *Δste7* strain in this condition (Fig 4B).

In addition, we determined alcohol dehydrogenase activity (ADH) in the protein kinase deletion strains, as the utilization of simpler alternative carbon sources, such as ethanol was also shown to be subject to control by CCR [5]. As expected, ADH activity was repressed in the presence of glucose whereas alcohol alone resulted in significantly higher ADH activity in the WT strain (Fig 4C). The same pattern of ADH activity was observed in the *Δste7* and *ΔmpkB* strains, although enzyme activity was significantly higher in the *Δste7* strain and significantly reduced in the *ΔmpkB* strain in the presence of ethanol (Fig 4C). In contrast, ADH activity remained high in the *ΔpbsA* strain in the control condition and in the presence of glucose.

In summary, these results suggest that MpkB is involved indirectly in CreA protein degradation, resulting in enzyme activities that are significantly reduced in the *ΔmpkB* strain, likely

**a**

## WT CreA-GFP

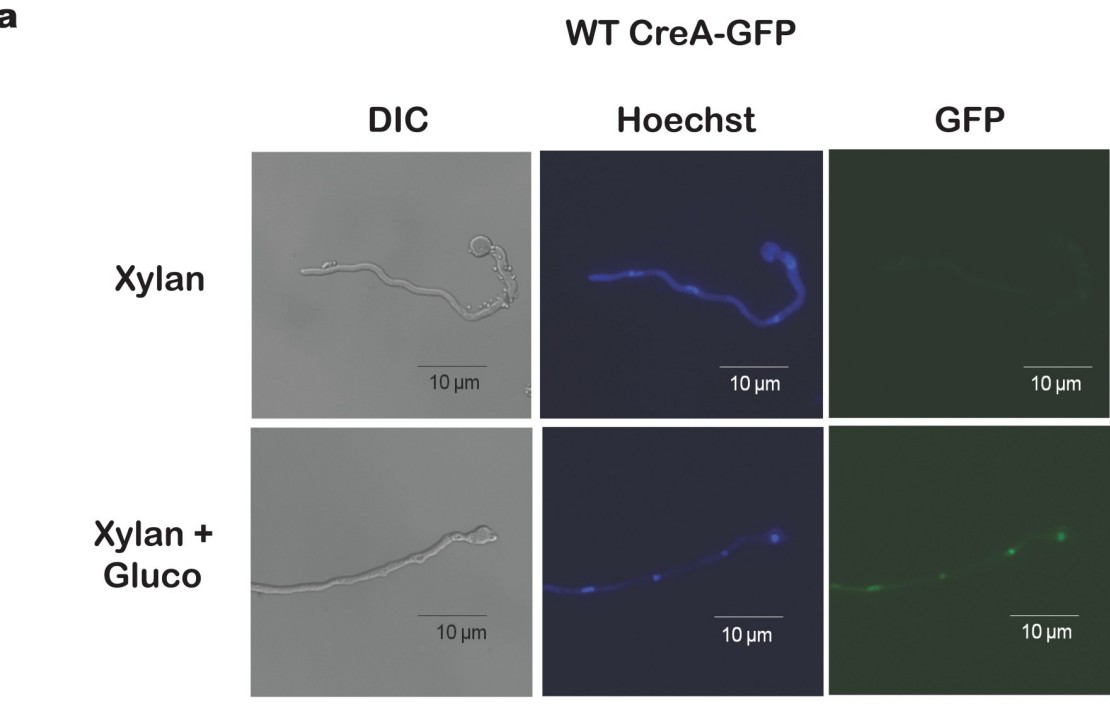

**b**

## Microscopy CreA-GFP

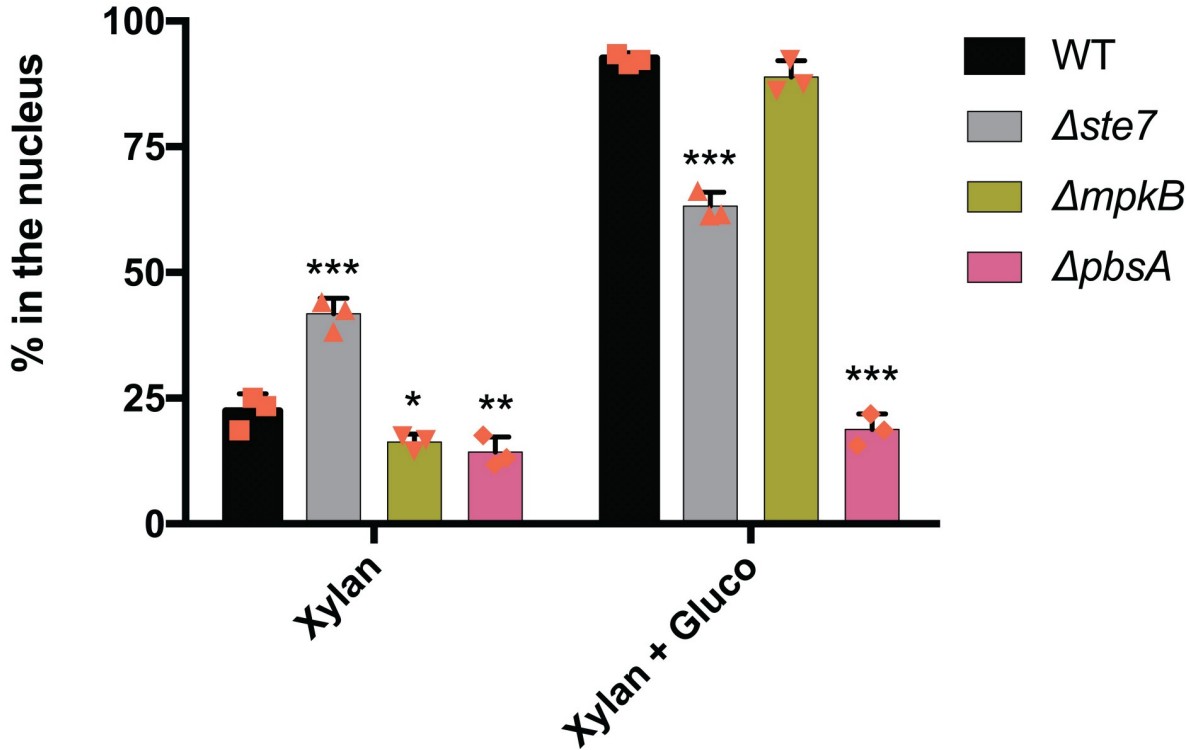

**Fig 3. Mitogen-activated protein kinases (MAPKs) are important for CreA cellular localization.** Microscopy pictures of wild-type (WT) hyphae, taken after 16 h growth at 22°C in xylan minimal medium (MM) and after the addition of glucose for 30 min, show localization of CreA-GFP to the nucleus. Pictures were taken at different wavelengths (DIC = differential interference contrast, Hoechst = Hoechst 33258 nucleic acid stain and GFP = green fluorescent protein), scale bar showed on the bottom (a). Percentage of CreA-GFP nuclear localization in different strains. Strains were grown as specified in (a) before nuclei with and without GFP were counted for 100 hyphal germlings for each condition and the % of CreA-GFP localization was calculated. Hyphae were stained with Hoechst 33258 in order to confirm GFP nuclear localization (b).

due to the presence of CreA in CC-de-repressing conditions. In contrast, PbsA is required for CreA protein biosynthesis under CCR condition, resulting in enzyme activities that are highly induced in the *ΔpbsA* strain, likely due to the absence of CreA in both CC-de-repressing and CCR conditions. Furthermore, it is possible that PbsA regulates CreA-GFP protein biosynthesis via the PKA pathway. Ste7 regulation of CreA and enzyme production is more complex, suggesting that Ste7 regulates different pathways.

## Identification of Ste7, MpkB and PbsA protein interaction partners and phosphorylation sites

In order to decipher the regulatory roles and pathways of the MAPKs Ste7, MpkB and PbsA during CCR and CC-de-repressing conditions, the corresponding interaction partners and/or target proteins were identified in the presence of CC-de-repressing and CC-repressing conditions. We therefore constructed C-terminal GFP-tagged Ste7, MpkB and PbsA strains in the AGB551 background strain and used the same genetic construction to complement the corresponding deletion strains. Deletion strains constructed using the AGB551 wild-type strain had the same growth phenotype in the presence of 2DG and AA as the one observed for the *A. nidulans* NPK deletion library (Fig 1A and 1B). The GFP-tagged and -complemented strains showed identical growth to the wild-type strain in the presence of 2DG and AA, indicating that they were functional (S3A Fig). Furthermore, we were also able to detect full length Ste7-GFP, MpkB-GFP and PbsA-GFP by Western blot in total protein extracts of strains that were grown in CC-de-repressing and CCR conditions (S3B Fig), confirming the biosynthesis of the tagged proteins in *A. nidulans*.

To identify potential interaction and/or target proteins of Ste7, MpkB and PbsA in CC-de-repressing and CC-repressing conditions, GFP-pulldown experiments were performed followed by mass spectrometry (MS) analysis for protein identification (S1 Table). Total proteins were extracted from strains grown in the aforementioned conditions, the GFP-tagged proteins Ste7, MpkB and PbsA were immunoprecipitated and protein samples were digested with trypsin. As a control for unspecific binding/enrichment, pull-downs were also performed with a strain expressing free GFP. We considered proteins as putative interaction partners of the target proteins that were detected in all three biological replicates but not in the control carrying free GFP.

When strains were grown in the presence of xylan, 80, 16 and 13 unique putative interaction partners were identified for Ste7, MpkB and PbsA, respectively; whereas after the addition of glucose, 54, 9 and 13 unique putative interaction partners were identified for Ste7, MpkB and PbsA, respectively; with 8 and 5 proteins identified for all three protein kinases in the presence of xylan and glucose, respectively (Fig 5A). These results suggest that these MAPKs could form part of a protein complex. We then focused on Ste7, MpkB and PbsA interaction partners that are predicted to be involved in MAPK pathways (Fig 5B). Based on these results (Fig 5B), Ste7, MpkB and PbsA are predicted to form various intermediary and/or transient protein complexes. These complexes consist of MpkB, Ste7, and MsgA in the MpkB-GFP pull-down assay and of PbsA, SskB, and GskA when considering the PbsA-GFP pull-down assay in the presence of xylan. Finally, the Ste7-GFP immunoprecipitation assay shows possible formation

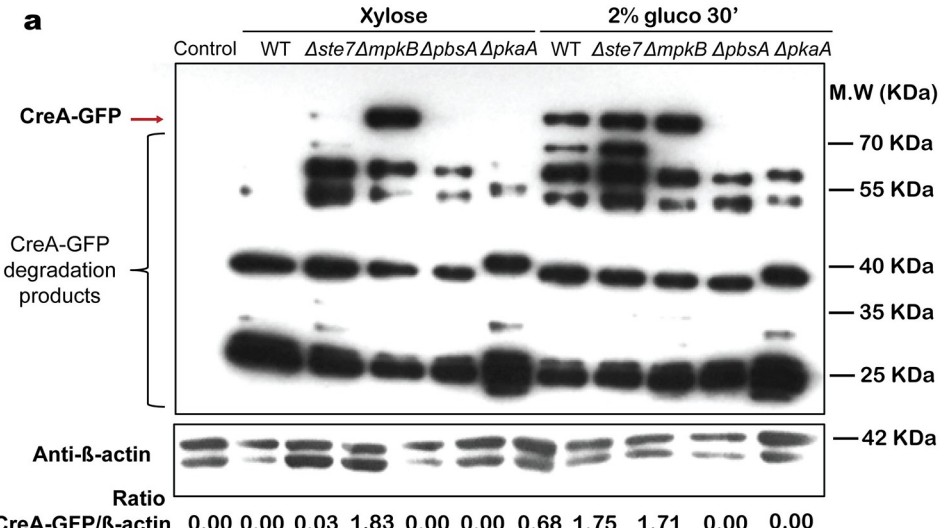

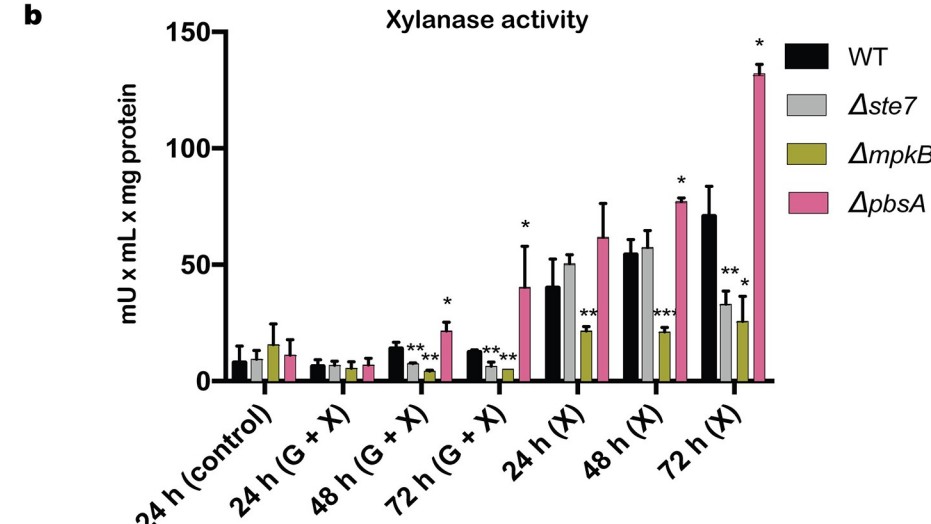

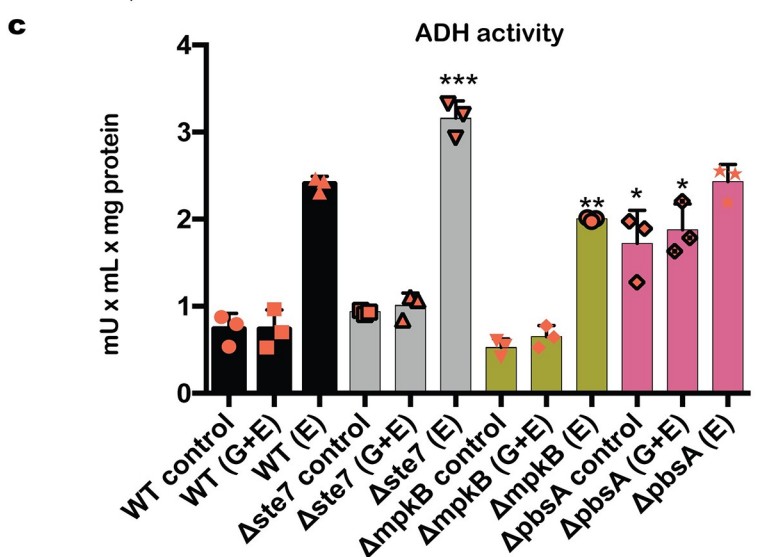

**Fig 4. Mitogen-activated protein kinases (MAPK) are important for CreA protein turnover and enzyme production.** The protein kinases are crucial for CreA-GFP protein turnover. The WT, *Δste7*, *ΔmpkB*, *ΔpbsA* and *ΔpkaA* strains were grown in minimal medium (MM) supplemented with 1% xylose (CC-de-repressing condition) for 24 h before 2% glucose (CCR) was added for 30 min and total cellular proteins were extracted. The GFP tagged proteins were pulled-down using GFP-trap beads and further analyzed by Western blot. The red arrow indicates full length CreA-GFP (76 KDa) and the bottom panel shows a Western blot where an antibody against β-actin was used in total cellular protein extracts that were used as input for immunoprecipitation. CreA-GFP/β-actin ratios were calculated using densitometric scans of full length CreA-GFP (a). Xylanase activity is impaired in the protein kinase deletion strains. The WT, *Δste7*, *ΔmpkB* and *ΔpbsA* strains were grown in MM supplemented with fructose (control) for 24 h before mycelia were transferred to a carbon catabolite (CC)-repressing condition (MM + 2% Glucose + 1% xylose) or CC-de-repressing condition (MM + 1% xylose) for 24, 48 and 72 h. Xylanase activity was measured in culture supernatants and normalized by dry weight (b). Alcohol dehydrogenase (ADH) activity is impaired in the protein kinase deletion strains. Strains were grown as described in (b) except that xylose was replaced by ethanol and incubation time was 2 h. ADH activity was measured in mycelial protein extracts and normalized by total intracellular protein (c). Standard deviations (b and c) represent the average of three biological replicates (shown as orange symbols) with $^*p<0.05$, $^{**}p<0.01$ and $^{***}p<0.001$ in a two-way ANOVA multiple comparison test.

of a large complex consisting of NikA, Ste7, MpkB, MsgA, SteC, SteD, SskB, SskA, FphA, MpkA, SakA, PbsA and GskA in the presence of xylan (Fig 5B). Upon the addition of glucose, the composition of these complexes changes, with only one protein interaction partner being different in the MpkB-GFP and PbsA-GFP pull-down assays when compared to the xylan condition (Fig 5B). In contrast, dissociation of the large protein complex identified during the Ste7-GFP pull-down assay is predicted to occur in the presence of glucose. Dissociation led to the formation of smaller complexes: a) the first is composed of NikA, Hk8, Ste7, MpkB, SteC, SteD, HamE and MsgA; b) the second comprises MpkB, Ste7, MsgA, and HamE; and c) the third complex is composed by Ste7, GskA, and PbsA (Fig 5B). In addition, the number of fragment spectra (independent scans) identified on Ste7, indicates increased phosphorylation in de-repressing conditions, whereas increased phosphorylation on PbsA was observed in conditions of CCR (Fig 5C). No difference in phosphorylation was observed for MpkB. When analyzing PbsA, putative phosphorylation sites are located within two regions, with one region being localized outside the HOG1-binding domain (HBD) and the kinase domain, and the other region being within the HBD (Fig 5D). Two hot spots for PbsA phosphorylation were observed and these may play a role in regulation. Phospho-site prediction shows that all phospho-sites identified on PbsA by MS are predicted to be targeted by glycogen synthase kinase A (GskA) (Fig 5D).

## GskA is important for SakA phosphorylation, PKA and xylanase activities

Of particular interest was the identification of GskA as a PbsA interaction partner in the presence of CC-de-repressing and CC-repressing conditions in our MS data. Furthermore, we identified several phosphorylation sites on PbsA using MS; *in silico* analysis of these sites predicts GskA to be the protein kinase which phosphorylates PbsA in these conditions. To further determine and confirm the regulation of PbsA by GskA, we performed several experiments with the GskA inhibitor Gsk3β inhibitor VII (*IgskA*), as deletion of *gskA* results in a strain that is severely growth compromised. Similar Gsk3 inhibitor compounds have already been shown as inhibiting *A. nidulans* growth [50]. First of all, the minimal inhibitory concentration (MIC) of *IgskA* was determined. In the presence of 30 μM *IgskA*, growth of all strains was inhibited except for the *Δste7* strain, which was resistant (Fig 6A). Furthermore, the three deletion strains were significantly more sensitive to 10 μM *IgskA* than when compared to the WT strain (Fig 6A) and we used this sub-inhibitory concentration for further experiments.

We showed that PbsA is important for the regulation of SakA phosphorylation (Fig 2B), PKA activity (Fig 2C) and xylanase production (Fig 4B) in *A. nidulans* and in *A. fumigatus*,

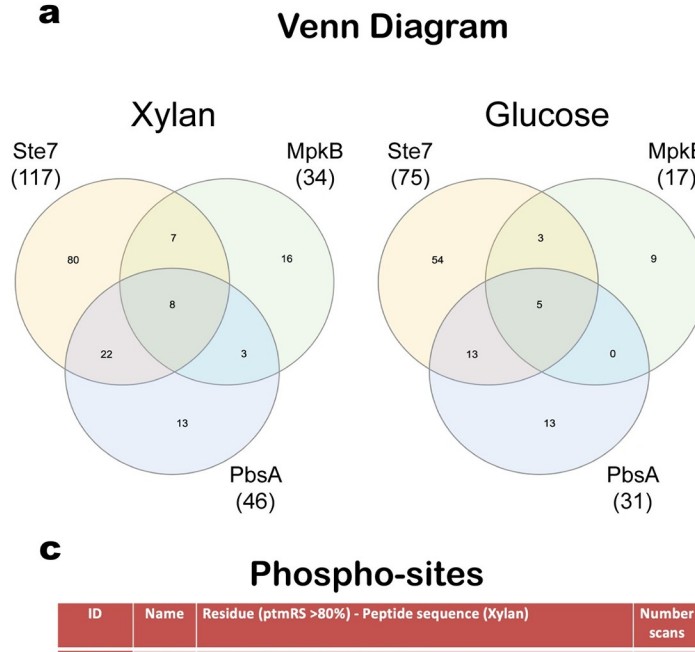

## a — Venn Diagram

**Xylan**

Ste7 (117), MpkB (34), PbsA (46)
80, 7, 16, 8, 22, 3, 13

**Glucose**

Ste7 (75), MpkB (17), PbsA (31)
54, 3, 9, 5, 13, 0, 13

## b — Summary MAPK interactions

| Selected proteins | GFP pulldow - Xylan | | | GFP pulldow - Glucose | | |
|---|---|---|---|---|---|---|
| | Ste7 | MpkB | PbsA | Ste7 | MpkB | PbsA |
| NikA | x | | | x | | |
| Ste7 | x | x | | x | x | x |
| MpkB | x | x | | x | x | |
| SteC | x | | | x | | |
| SteD | x | | | x | | |
| SskB | x | | x | | | |
| SskA | x | | | | | |
| GskA | x | | x | | | x |
| PbsA | x | | x | | | x |
| MpkA | x | | | | | |
| FphA | x | | | | | |
| SakA | x | | | | | |
| MsgA | x | X | | | x | |
| HamE | | | | x | x | |
| HK8 | | | | x | | |

## c — Phospho-sites

| ID | Name | Residue (ptmRS >80%) - Peptide sequence (Xylan) | Number scans |
|---|---|---|---|
| AN3422 | Ste7 | S445 (81.7) - SSRsSPPISLEHLSLESK | 1 |
| | | S376 (97.5) - SLKETREsPSPAQAPSPVQK | 3 |
| | | S384 (99.6) - ESPSPAQAPsPVQK | 1 |
| AN3719 | MpkB | ND | ND |
| AN0931 | PbsA | S15 (85.0) - SAEHDFAAENPPVsPHsDGDsAPVTLEDEVSsPNTK | ND |
| | | S22 (94.0) - SAEHDFAAENPPVsPHsDGDsAPVTLEDEVSsPNTK | 1 |
| | | S174 (100) - NWASAPTVGGGsPVsGsPKGGLAAKR | 1 |
| | | S174 (100) - LPPTHRPGPPKNWASAPTVGGGsPVsGsPK | 1 |
| | | S174 (100) - NWASAPTVGGGsPVsGsPK | 5 |
| | | S177 (87.0) - NWASAPTVGGGsPVsGsPKGGLAAKR | 1 |
| | | S179 (88.0) - NWASAPTVGGGsPVsGsPKGGLAAKR | 1 |
| | | S179 (100) - NWASAPTVGGGsPVsGsPK | 5 |

| ID | Name | Residue (ptmRS >80%) - Peptide sequence (Glucose) | Number scans |
|---|---|---|---|
| AN3422 | Ste7 | ND | ND |
| AN3719 | MpkB | ND | ND |
| AN0931 | PbsA | S15 (91.0) - SAEHDFAAENPPVsPHsDGDsAPVTLEDEVSsPNTK | 2 |
| | | S18 (96.0) - SAEHDFAAENPPVsPHsDGDsAPVTLEDEVSsPNTK | 1 |
| | | S22 (94.0) - SAEHDFAAENPPVsPHsDGDsAPVTLEDEVSsPNTK | 1 |
| | | S174 (100) - NWASAPTVGGGsPVsGsPKGGLAAKR | 2 |
| | | S174 (100) - NWASAPTVGGGsPVsGsPK | 4 |
| | | S174 (100) - LPPTHRPGPPKNWASAPTVGGGsPVsGsPK | 4 |
| | | S177 (87.0) - NWASAPTVGGGsPVsGsPKGGLAAKR | 2 |
| | | S179 (88.0) - NWASAPTVGGGsPVsGsPKGGLAAKR | 2 |
| | | S179 (100) - NWASAPTVGGGsPVsGsPK | 4 |

## d — Phosphorylations on PbsA

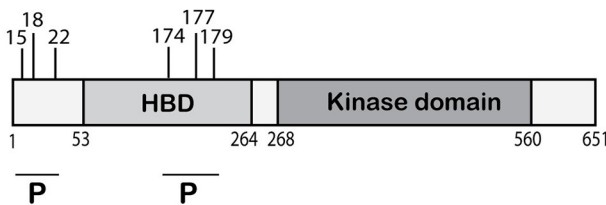

18, 15, 22, 177, 174, 179

HBD — Kinase domain

1, 53, 264, 268, 560, 651

P    P

### GSK3 Predition phospho-sites

| Position | Score |
|---|---|
| S15 | 0.50 |
| S18 | 0.47 |
| S22 | 0.46 |
| S174 | 0.49 |
| S177 | 0.45 |
| S179 | 0.47 |

**Fig 5. Identification of mitogen-activated protein kinase (MAPK) protein interactions and phosphorylation sites.** Venn diagram depicting the number of proteins, identified by mass spectrometry (MS), that potentially interact with Ste7-GFP, MpkB-GFP and PbsA-GFP in the presence of xylan or xylan and glucose. Immunoprecipitation assays were carried out for the Ste7-GFP, MpkB-GFP and PbsA-GFP strains and samples were analyzed by MS/MS. (a). Table summarizing the interaction of Ste7-GFP, MpkB-GFP and PbsA-GFP with proteins that are predicted to participate in MAPK pathways in the presence of xylan or xylan and glucose. Crosses indicate a putative interaction whereas the absence of a cross signifies that the protein was not identified by MS (b). Phosphorylation sites on Ste7, MpkB and PbsA identified by phospho-proteomics in the presence of CC-de-repressing (red, xylan) and CC-repressing (blue, xylan+glucose) conditions. Identified phosphorylation sites had a statistical analysis for a putative modification higher than 80% (ptmRS> 80%), single phosphorylation sites in peptides are shown in red and the quantification by number of identified scans (c). Phosphorylation sites on PbsA (1-651aa), HOG1-binding domain (HBD, 53-264aa) and kinase domain (268-560aa), P shows the regions of phosphorylation on PbsA, bottom shows the prediction of GSK3 phospho-site on PbsA (d).

PKA activity is regulated by SakA in osmotic stress conditions [28]. To further confirm the role of GskA in PbsA regulation, we measured PKA activity in the WT as well as in WT cultures treated with the GSK inhibitor. Addition of the GSK inhibitor resulted in significantly higher PKA activity in CC-repressing conditions only (Fig 6B). To further determine a role of

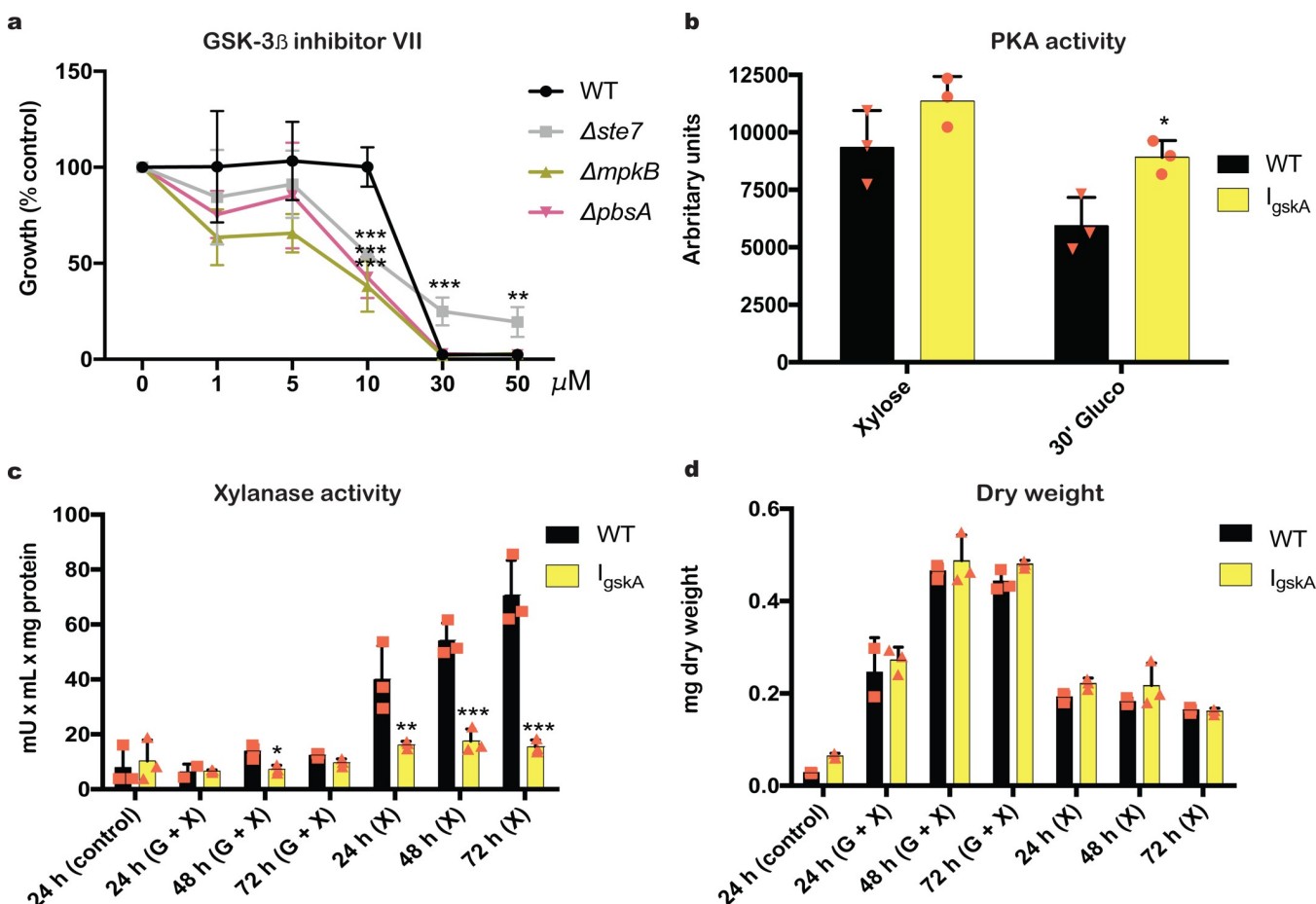

**Fig 6. Glycogen synthase kinase (GskA) is important for protein kinase A (PKA) and xylanase activities.** The GSK3β inhibitor VII inhibited growth of all strains at 30 μM. The WT, *Δste7*, *ΔmpkB* and *ΔpbsA* strains were grown in biological triplicates in 96-well plates for 48 h in glucose minimal medium (MM) supplemented with increasing concentrations of the GSK3β inhibitor VII before O.D.600nm (optical density) was measured and the percentage of growth was calculated as a reference to the control, drug-free condition (a). PKA activity is increased in the presence of the GSK3β inhibitor VII. Strains were grown in xylose MM for 24 h before mycelia were transferred to ddH₂O with or without 10 μM GSK3β inhibitor VII (*IgskA*) for 2 h. Subsequently mycelia were transferred to xylose MM or xylose plus 2% glucose MM with or without *IgskA* for 30 min. PKA activity was measured in 10 μg of total extracted intracellular protein (b). GSK inhibition reduces xylanase activity in the presence of xylose. Xylanase activity was measured in culture supernatants of strains grown for 24 in fructose MM (control) and mycelia were transferred to ddH₂O with or without 10 μM GSK3β inhibitor VII (*IgskA*) for 2 h. Subsequently mycelia were transfer to MM supplemented with glucose and xylose or xylose only in the presence of *IgskA* for 24, 48 and 72 h (c). Addition of the Gsk-3β inhibitor VII does not impair fungal growth. Fungal dry weight of strains grown in the same conditions as specified in c (d). Standard deviations represent the average of three biological replicates (shown as orange symbols) with *p<0.05, **p<0.01 and ***p<0.001 in a two-way ANOVA multiple comparison test.

GSK inhibition in CCR, xylanase activity was measured in culture supernatants of the WT strain in the absence and presence of *IgskA*, when grown for 24 h in the control condition and after transfer to CC-de-repressing and CC-repressing conditions for 24 h, 48 h and 72 h (Fig 6C). Addition of *IgskA* significantly reduced xylanase activity in culture supernatants in CC-de-repressing conditions, suggesting that GSK activity is important for enzyme production (Fig 6C). Xylanase activities were normalized by fungal dry weight, which was not significantly different in the presence of the GSK inhibitor (Fig 6D), suggesting that the concentration of *IgskA* did not affect fungal growth after pre-growth in control condition. These results are in contrast to the results on PKA and xylanase activities obtained for the *ΔpbsA* (Fig 2C) strain, and suggest that either GSK-mediated regulation of PbsA does not mimic a deletion pheno-type for enzyme activity, or that GSK is important for enzyme activity either through direct

regulation or through controlling other protein kinases. In summary, these results suggest that GskA is important for the regulation of PKA and xylanase activities in CC-de-repressing and CC-repressing conditions.

### The PbsA phosphorylation sites are important for osmotic stress resistance, HOG pathway activation and enzyme activities

Next, we further determined the function of the two putative PbsA phosphorylation sites that are predicted to be targeted by GskA (Fig 5D). A strain was constructed where PbsA serine 22 and serine 179 were mutated to alanine in order to mimic continued absence of phosphorylation. The PbsA$^{S22A\ S179A}$ strain was significantly more sensitive to sorbitol-induced osmotic stress, as shown by a reduction in radial growth (Fig 7A), suggesting impairment in the HOG pathway. Furthermore, the PbsA$^{S22A\ S179A}$ strain was significantly more sensitive to 2DG (Fig 7B) and resistant to AA (Fig 7C), which is the reverse phenotype than the one observed for the *ΔpbsA* strain (Fig 1A and 1B) and suggests that these two sites are important for the regulatory function of PbsA. Indeed, SakA phosphorylation, as determined by Western blot, was significantly reduced in the PbsA$^{S22A\ S179A}$ strain when compared to the WT strain (Fig 7D), confirming an inactivation of the HOG pathway in these conditions. Next, we determined the effect of GSK inhibition through the addition of *lgskA* on HOG pathway activation by performing Western blot analysis of SakA phosphorylation in both CC-de-repressing and CC-repressing conditions. Addition of *IgskA* resulted in the complete absence of SakA phosphorylation in all conditions, suggesting a putative interaction between GskA and the HOG pathway (Fig 7D). In addition, both PKA (Fig 7E) and xylanase (Fig 7F) activities were significantly reduced in the PbsA$^{S22A\ S179A}$ strain in both CC-de-repressing and CC-repressing conditions. These findings are in line with the aforementioned GskA inhibition studies and suggest serines S22 and S179 as target sites of GskA in *A. nidulans*. In summary, we show that the PbsA phosphorylation sites are important for osmotic stress resistance, HOG pathway activation and activities of enzymes that respond to carbon source sensing and regulation.

## Discussion

CCR is an extremely complex process that is subject to various regulatory pathways and involves post-translational modifications, such as phosphorylation, on signaling pathways and effector proteins in a variety of fungi [11,15,23,51]. In the present study, we screened a NPK deletion library for defects in CCR, which led to the identification of several MAPK pathway-associated protein kinases as important for CCR, and we selected Ste7, MpkB and PbsA for further characterization.

We show that growth of the wild-type strain in the presence of alternative, non-glucose carbon sources activates the HOG MAPK pathway, decreases CreA protein levels and increases xylanase and PKA activities; whereas the presence of glucose inactivates the HOG MAPK pathway and increases CreA protein levels as well as significantly decreasing xylanase and PKA activities (S4 Fig). In contrast, SakA was not phosphorylated, CreA was not observed in the nucleus, CreA protein levels were not detected and enzyme activities were highly increased in both CC-de-repressing and CCR conditions in the *ΔpbsA* strain (S4 Fig). These results suggest that PbsA is crucial for the stability and/or the biosynthesis of CreA protein levels under CCR conditions, and reinforce that PbsA-mediated phosphorylation of SakA is required for the regulation of CCR. In CC-de-repressing conditions, SakA is phosphorylated and SakA-GFP is predominantly nuclear; whereas the addition of glucose causes a reduction in SakA phosphorylation and translocation to the cytoplasm, suggesting a role for SakA in the utilization of non-preferred carbon sources and de-repressing phenotype (S2B Fig). Mutation of the two PbsA

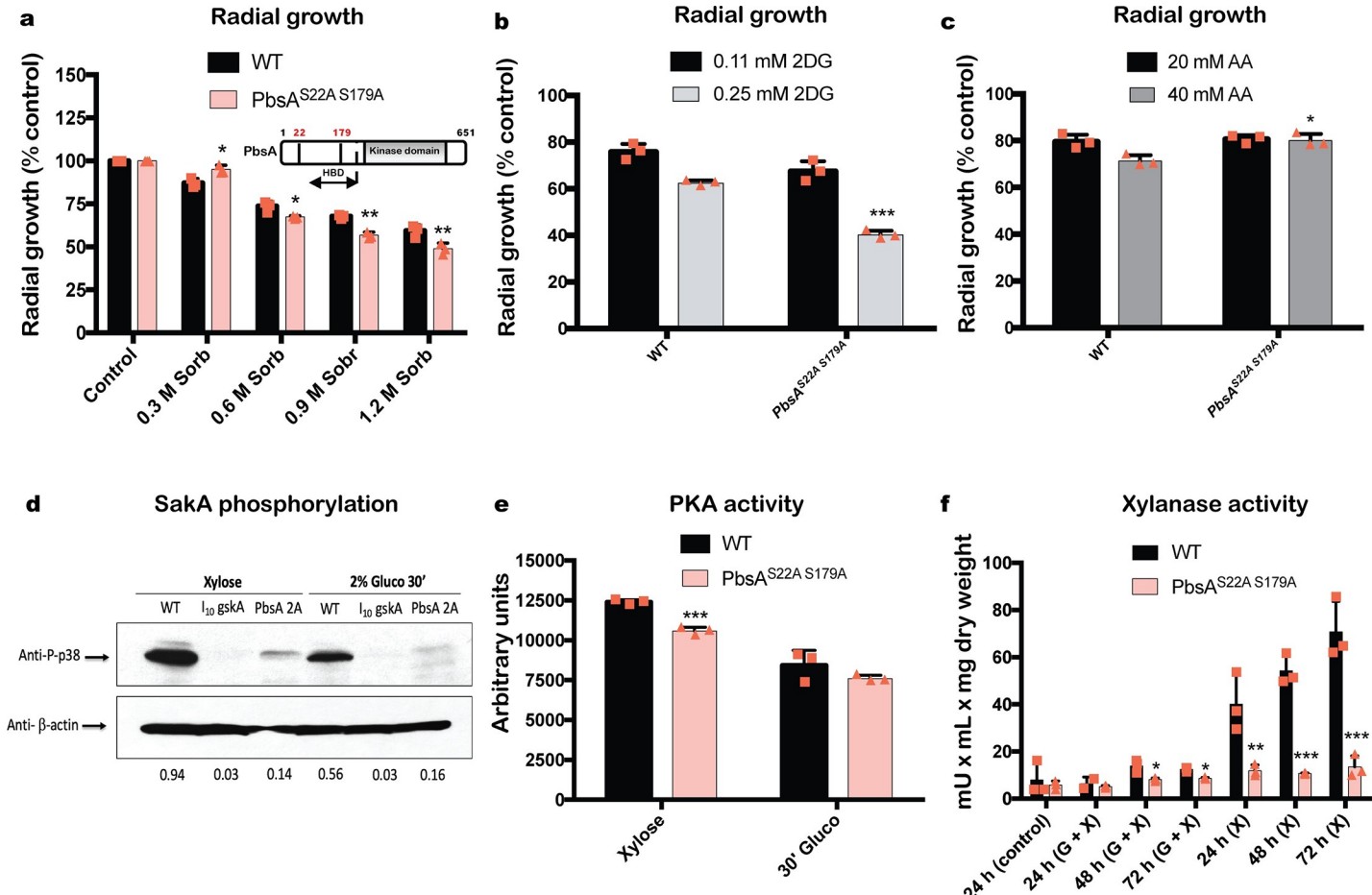

**Fig 7. The PbsA phosphorylation sites are important for osmotic stress resistance, HOG pathway activation and enzyme activities.** The PbsA^S22A S179A strain is sensitive to sorbitol-induced osmotic stress. Strains were grown from 10^5 spores in glucose MM 0.3, 0.6, 0.9 and 1.2 M sorbitol for 5 days at 37°C before colony radial diameter was measured and the percentage of growth in comparison to the control (no sorbitol) condition was calculated. Also shown is a diagram that depicts the localization of the PbsA double point mutations (HBD = Hog1 binding domain) (a). The PbsA^S22A S179A strain is sensitive to 2-deoxyglucose (2DG) (b) and resistant to allyl alcohol (AA) (c). Strains were grown and growth was calculated as specified in (a), except that xylose MM was used for the 2DG assays. SakA phosphorylation is significantly reduced in the presence of 10 μM of the Gsk-3β inhibitor VII (l_10gskA) and in the PbsA^S22A S179A strain. SakA phosphorylation was determined by Western blot, using an anti-P-p38 antibody, in total cellular protein extracts of strains grown for 24 h in xylose MM and after the addition of glucose for 30 min. Treatment of WT cultures with the Gsk-3β inhibitor VII was carried out as described in Fig 5 and SakA phosphorylation levels were normalized by the anti-β-actin antibody, with the bottom panel indicating the P-p38/actin ratio (d). PKA (e) activity and extracellular xylanase (f) activities were reduced in the PbsA^S22A S179A strain. PKA activity was measured in total cellular protein extracts of strains grown in the same conditions as specified in Fig 2 (e). Xylanase activity was measured in culture supernatants of strains grown for 24 h in fructose MM (control) and after transfer to glucose and xylose (G+X) or xylose (X) MM for 24 h, 48 h and 72 h (f). Standard deviations represent the average of three biological replicates (shown as orange symbols) with *p<0.05, **p<0.01 and ***p<0.001 in a two-way ANOVA multiple comparison test.

serines 22 and 179 to alanines, predicted to be phosphorylated by GskA, and addition of the GskA inhibitor lgskA, impaired SakA phosphorylation, resulting in a CC-repressing phenotype, further supporting a role for PbsA and SakA in CCR (S4 Fig).

In the *Δste7* and *ΔmpkB* strains, SakA phosphorylation, CreA cellular localization and protein stability as well as PKA and xylanase activities were also impaired (S4 Fig), suggesting that CC-de-repression requires activation of the HOG and PKA pathways at the same time. Ste7 and MpkB are part of the same MAPK pathway but HOG pathway activation, PKA activity and defects in CCR differ between both deletion strains, suggesting that they control these processes through different mechanisms. This is further substantiated through additional

predicted interactions with other protein kinases, such as SnfA (Fig 2A), which has been shown to be crucial for CCR and CreA localization [1].

In addition, the PKA signaling pathway is also important for CCR and potentially interacts with the HOG pathway. In both the *ΔpbsA* and *ΔpkaA* strains, CreA protein levels could not be detected, although microscopy shows GFP fluorescence in the nucleus in both strains. This discrepancy may be explained by residual CreA-GFP degradation products still being present in the nucleus (Fig 4A). Regulation of CreA by the HOG and PKA pathways could either occur in parallel, or PkaA and SakA interact, suggesting cross-talk between both pathways. In *A. fumigatus*, SakA and PkaC1 were shown to physically interact in osmotic stress conditions [28]. In *A. nidulans*, deletion of *pbsA* results in very low levels of SakA phosphorylation and PKA activity in CC-de-repressing and CC-repressing conditions, suggesting that like in *A. fumigatus*, cross-talk between the HOG and PKA pathways exist in *A. nidulans*. Furthermore, we cannot exclude additional regulatory mechanisms acting on PKA signaling, exerted by Ste7 and MpkB, and which could occur either through the HOG pathway or through another, uncharacterized signaling route. Nevertheless, the aforementioned results are strong evidence that support cross-talk between the HOG, PKA and CCR pathways and highlight the complexity of signaling input required for carbon source utilization.

In order to gain mechanistic insights into Ste7, MpkB and PbsA-mediated regulation of CCR, MS was carried out to identify interaction partners. Furthermore, phospho-proteomic analyses identified potential phosphorylation sites on two MAPKs. Of particular interest was the identification of GskA as an interaction partner for PbsA in all conditions, as well as indication of phosphorylation events on PbsA, that are predicted (NetPhos3.1) [52,53] to be catalyzed by GskA. A previous study showed that GskA interacts with CreA in the presence of xylan, whereas upon the addition of glucose, this interaction is lost, with CreA moving to the nucleus with the co-repressors SsnF and RcoA, and GskA remaining cytoplasmic [7]. Pharmacological inhibition of GskA, resulted in significantly reduced xylanase activity (Fig 6C) and the absence of phosphorylated SakA (Fig 7D), further supporting an interaction between GskA and the HOG pathway. Similar results were obtained for PbsA$^{S22A\ S179A}$ mutant, suggesting that GskA is the protein kinase that targets these sites. In *S. cerevisiae*, the Pbs2p Hog1-binding domain (HBD) region, that is required for Pbs2p binding to Hog1p, is located between amino acids 136–245 [54]. It is tempting to suggest a similar role for serine 179 of *A. nidulans* PbsA (Fig 7A), especially as the PbsA$^{S22A\ S179A}$ strain had reduced SakA phosphorylation levels (Fig 7D) and increased sensitivity to osmotic stress (Fig 7A), thus suggesting that the phosphomutations impact PbsA function. Alternatively, as mutation of the two PbsA phosphorylation sites had an effect on PKA activity (Fig 7E), this could be responsible for the observed phenotypes. On the other hand, PKA activity was affected by GSK inhibition in CC-repressing conditions, although SakA phosphorylation was almost undetectable in the presence of the GSK inhibitor and PbsA$^{S22A\ S179A}$ strain, suggesting that additional pathways exist to regulate PKA enzyme activity.

Based on the aforementioned results, our MS analysis and previous studies [7,11,15], we propose a model for the interaction between the different signaling pathways that ultimately control CCR (Fig 8). In the presence of the alternative carbon source xylan, formation of transient protein complexes is observed that change upon the addition of glucose. Notably, the Ste7-GFP pull-down assay identified the proteins NikA, Ste7, SteC, SteD, MsgA, MpkB, SskA, SskB, FphA, MpkA, GskA, PbsA and SakA as forming a complex, with Ste7 and SakA being phosphorylated, resulting in PKA pathway activation which is required for CreA nuclear localization. Additional signals are also predicted to keep CreA from entering the nucleus, with SakA possibly involved in this process, and which together control a dynamic cytoplasm/ nucleus shuttling of CreA under CC-de-repressing conditions (Fig 8A). NikA is a histidine

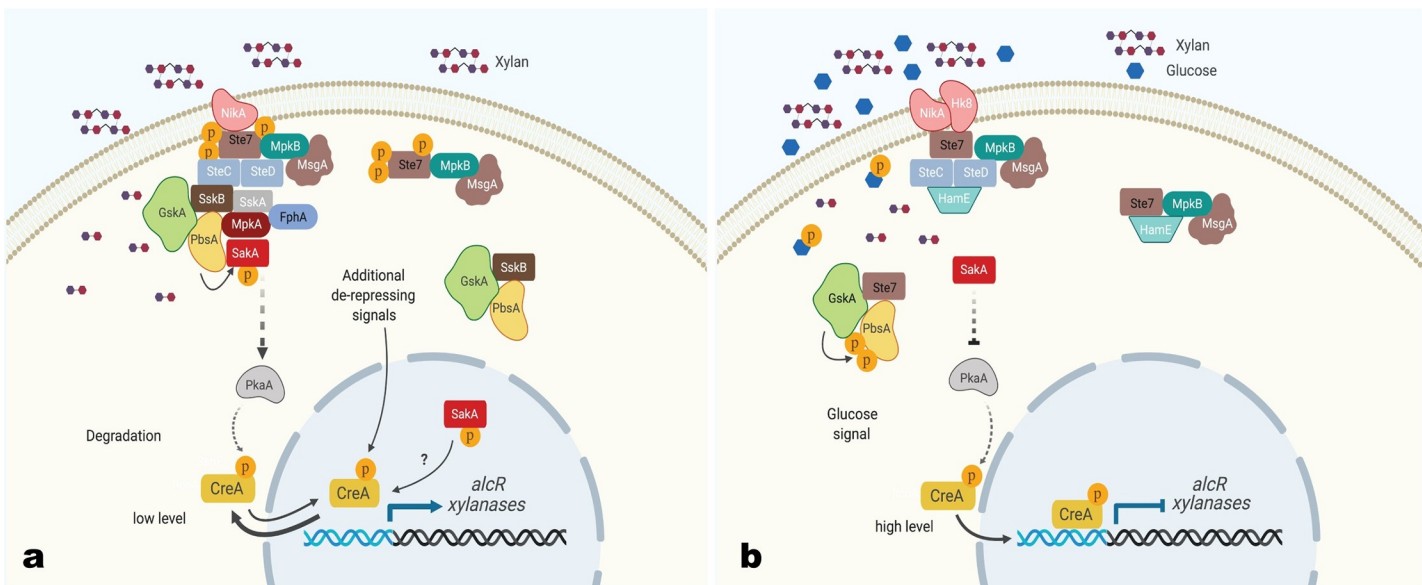

**Fig 8. Diagram depicting a potential interaction of mitogen-activated protein kinase (MAPK) and protein kinase A (PKA) pathways in the regulation of CreA in carbon catabolite de-repressing (xylan) and repressing (xylan and glucose) conditions.** In the presence of xylan, transient complexes are formed including a large protein complex that was observed in Ste7-GFP pull-down assay. The large complex consists of the NikA, Ste7, SteC, SteD, MsgA, MpkB, SskA, SskB, FphA, MpkA, GskA, PbsA and SakA proteins, resulting in SakA phosphorylation and HOG (high osmolarity glycerol) pathway activation. Phosphorylated SakA is mainly nuclear. SakA phosphorylation may occur i) either through the phytochrome photoreceptor FphA which activates the SskA-SskB-PbsA signaling cascade and/or ii) through the histidine-specific protein kinase NikA which relays a signal to SteC-SteD, resulting in PbsA-mediated activation of SakA. In addition, this complex regulates and increases PKA activity which in turn controls cellular localization of the CreA repressor complex. In the presence of xylan, the CreA-repressor complex has low levels and it is mainly cytoplasmic with a small percentage also observed in the nucleus. CreA-repressor complex localization is therefore regulated by the HOG and PKA pathways as well as additional, unknown signals in CC-de-repressing conditions which cause its nuclear exclusion and degradation (a). Upon the addition of glucose, the MAPK dissociates into smaller protein complexes, likely resulting in SakA de-phosphorylation and HOG pathway inactivation. Complex dissociation may be caused by i) either Ste7 de-phosphorylation by the phosphatase MsgA and/or ii) phosphorylation of PbsA by the glycogen kinase synthase GskA, resulting in a weakening of PbsA-SakA interaction. Inactivation of PbsA and subsequent decrease in SakA phosphorylation results in decreased PKA activity and constitutes a signal for CreA repressor-complex translocation to the nucleus and repression of target genes such as those encoding xylanases (b). Protein names are indicated in each shape; xylan is represented by polymers in purple and red and glucose by blue hexagons; P in orange squares represent phosphorylation events; arrows show the direction of phosphorylation with solid arrows representing direct phosphorylation and dashed arrows representing indirect phosphorylation and/or regulation. Membrane lipid bilayer and cytoplasm are depicted in light brown and nuclei are shown in light blue.

kinase that is part of a two-component signal transduction phospho-relay system [55,56], which together with the phytochrome photoreceptor FphA could trigger a MAPK phosphorylation cascade that ultimately results in SakA phosphorylation in these conditions. Indeed, FphA has previously been shown to cause SakA phosphorylation upon the sensing of light [45,57]. HOG MAPK pathway activation via SakA phosphorylation in the presence of CC-de-repressing conditions may therefore be mediated by a two-component system, as previously described for other signaling pathways [55].

Upon the addition of glucose, the HOG activation complex dissociates into smaller protein complexes: i) the first is composed of NikA, Hk8, Ste7, MpkB, SteC, SteD, HamE and the phosphatase MsgA, with Ste7 residues not being phosphorylated; as well as ii) additional transient protein interactions such as the complex composed of GskA, PbsA and SakA (Fig 8B), resulting in a decrease in SakA phosphorylation and reduced PKA pathway activation. It is tempting to speculate that MsgA may be responsible for Ste7 de-phosphorylation, subsequent protein complex dissociation and inactivation of the HOG pathway, as well as implementation of CCR. Indeed, the Δ*msgA* strain was not able to grow in the presence of glucose as sole carbon source [58], although future studies are required to determine the role of this phosphatase in CCR. Together these signals then promote CreA repressor complex translocation to the

nucleus where it can repress target genes (Fig 8B). In the presence of xylan, immunoprecipitation (IP) of Ste7-GFP identified PbsA as an interaction partner, but when IP of PbsA-GFP was carried out, Ste7 was not identified (Fig 5B). Similarly, in the presence of glucose, PbsA was not identified as an interaction partner of Ste7-GFP, whereas IP of PbsA-GFP identified Ste7 as an interaction partner. This discrepancy may be explained by the interaction dynamics where Ste7 and PbsA interaction is weak and/or transient in both conditions, in addition to the formation of two protein complexes in the presence of glucose which also favors a dissociation between Ste7 and PbsA.

In summary, this study unravels part of the extremely complex mechanism that underlies the regulation of preferred and alternative carbon source utilization in a reference filamentous fungus, and highlights the interaction that takes place between protein kinases of several very different signaling pathways.

## Methods

### Strains and culture conditions

The *A. nidulans* non-essential protein kinase (NPK) deletion library was kindly provided by Dr. Osmani[30,59]. Briefly, NPK-encoding genes were deleted in the *A. nidulans* SO451 (*pyrG89; wA3; argB2; ΔnkuAku70::argB pyroA4; sE15 nirA14 chaA1 fwA1*) background strain by replacing the target gene with the *A. fumigatus pyrG* marker gene. All strains were grown at 37˚C, except for experiments with allyl alcohol, which were performed at 30˚C. Strains were grown in either liquid (without agar) or solid (with 20 g/l agar) complete medium (CM) or minimal medium (MM) as described previously [15]. The strains that were used for Western blot analyses, enzyme activity assays, mass spectrometry and microscopy and point mutations were constructed in the *A. nidulans* AGB551 background strain as described previously [7]. The strain harboring a double point mutation in *pbsA*, where serines 22 and 179 were mutated to an alanine, was constructed by replacing the endogenous gene by the respective *pbsA*[S22A S179A]-*gfp* cassette. gDNA from the PbsA-GFP (5'UTR-*pbsA-gfp-AfpyrG*-3'UTR) strain was used as a template for subsequent PCR amplifications. The following three fragments were generated: a) 5'UTR-*pbsA* (partial sequence until the first point mutation), b) *pbsA* fragment carrying the two mutations S22A S179A (using primers containing the mutations and 20 bp flanking sequence) and c) *pbsA* (partial sequence after the second point mutation)-*gfp-AfpyrG*-3'UTR. The three fragments were recombined into one cassette using plasmid pRS426 and transformed into *Saccharomyces cerevisiae* strain SC9721 (MATα *his3-Δ200 URA3-52 leu2Δ1 lys2Δ202 trp1Δ63*) obtained from the Fungal Genetic Stock Center (FGSC). All plasmids were cloned into bacteria before full cassettes were amplified by PCR from extracted bacterial plasmid DNA (Qiagen Plasmid miniprep) and used for transformation in *Aspergillus nidulans* as previously described. The presence of the *pbsA* double point mutation was confirmed by DNA sequencing. To determine fungal biomass dry weight, strains were grown in liquid MM for the specified time points, before mycelia were harvested by vacuum filtration, freeze-dried and weighed. All experiments were carried out in biological triplicates unless otherwise specified. Strains used in this study are listed in S2 Table.

### Screening of the NPK deletion library for growth in the presence of 2-deoxyglucose (2DG) and allyl alcohol (AA)

Strains were grown from $10^7$ spores in MM supplemented with xylose and increasing concentrations of 2DG, or MM supplemented with glucose and increasing concentrations of AA for

48 h in 96-well plates. Plates were inspected visually for strains that had increased or decreased growth in any of the two compounds. 2DG and AA sensitivity/resistance of selected strains was confirmed by measuring colony radial diameter when grown from $10^5$ spores for 5 days on plates containing the aforementioned MM and 2DG/AA combinations.

### Detection of glucose in the supernatant

Strains were grown from $10^7$ conidia in 30 mL CM for 24 h before mycelia were washed twice with ddH$_2$O and transferred to MM supplemented with 1% glucose. Supernatant samples were collected at the specified time points and glucose concentrations were measured using the Glucose GOD-PAP Liquid Stable Mono-reagent kit (LaborLab Laboratories Ltd. Guarulhos, São Paulo, Brazil) according to manufacturer's instructions. The percentage of residual glucose in the supernatants at different times was calculated with reference to time point 0.

### Microscopy

Strains were grown for 16 h at 22°C in 5 ml MM supplemented with 1% w/v xylan in small petri dishes containing a cover slip before glucose was added to a final concentration of 2% v/v and samples were incubated an additional 30 min. Cover slips containing the attached hyphal germlings were viewed under a Carl Zeiss (Jena, Germany) AxioObserver.Z1 fluorescent microscope using the 100x magnification oil immersion objective (EC Plan-Neofluar, NA 1.3). Differential interference contrast (DIC) and fluorescent images were taken and processed, and the percentage of nuclear CreA-GFP was calculated as described before [31].

### Protein kinase A (PKA) activity

Total cellular protein extraction and PKA activity was measured in 10 μg of total protein lysate using the Pep-Tag assay for non-radioactive detection of cAMP-dependent protein kinase kit (Promega), according to the manufacturer's instructions. The PKA phosphorylated substrate in samples was quantified by densitometry quantification using ImageJ and compared to the positive control (purified PKA catalytic subunit protein, 100% activity).

### Xylanase activity

Xylanase (endo-1,4-β-xylanase) activities were measured in culture supernatants using Azo-Xylan (Birch-wood, Megazyme) according to manufacturer's instructions.

### Alcohol dehydrogenase (ADH) activity

Total cellular protein extracts from mycelia grown in the specified conditions were obtained by re-suspending ground mycelia in 1 ml B250 buffer (250 mM NaCl, 100 mM Tris-HCl pH 7.5, 10% glycerol, 1 mM EDTA and 0.1% NP-40) supplemented with 1.5 ml/L of 1 M DTT, 1 pill/10mL of the Complete-mini Protease Inhibitor Cocktail EDTA-free (Roche), 3 ml/L of 0.5 M Benzamidine, 1 pill/10mL of phosphoSTOP phosphatase inhibitors and 10 ml/L of 100 mM PMSF and subsequent centrifugation for 10 min at 4°C, 13,200 rpm. ADH activity was measured in 10 μg of total intracellular protein lysate in a 96-well plate. Protein samples were re-suspended in reaction buffer (50 mM sodium pyrophosphate decahydrate, 50 μM semicarbazide hydrochloride and 20 mM glycine in pH 8.0), 6 mM NAD$^+$ and water to a final volume of 200 μL/well. Absorbance was read at 340 nm for 15 min at 37°C, with readings at every minute, using the Synergy HTX (BioTek) plate reader. Enzyme activity was calculated using the linear part of the slope and expressed as mU x mL x mg protein.

## Protein extraction and Western blot

Protein extractions were performed as described above and Western blots were carried out as described previously [7].

## GFP immunoprecipitation (IP) assays

Total cellular proteins were extracted as described above and supernatants were transferred to a new e-cup Eppendorf and kept on ice. Subsequently, 20 uL/sample of GFP-trap beads (Chromotek) were equilibrated in 0.5 mL B250 lysis buffer for 10 min on ice and beads were collected by centrifugation at 3,000 rpm for 5 min. Beads were then incubated with 6 mg of total protein lysate at 4°C for 3 h before samples were centrifuged and supernatants were discarded. The GFP-trap beads were washed twice using 1 mL B250 lysis buffer without DTT and one additional wash step was done with B250 lysis buffer containing DTT. GFP-trap beads were collected by centrifugation and supernatants were removed.

## Mass spectrometry analysis with nanoLC-nanoESI-MS/MS2

Nano LC- RSLCnano Ultimate 3000 system (Thermo Scientific): Peptides of 3 µl sample solution were loaded and washed on an Acclaim PepMap 100 column (75 µm x 2 cm, C18, 3 µm, 100 Å, Thermo Scientific) at a flow rate of 25 µl/min for 6 min in 100% solvent A (98% water, 2% acetonitrile, 0.07% TFA). Analytical peptide separation by reverse phase chromatography was performed on an Acclaim PepMap RSLC column (75 µm x 25 cm, C18, 3 µm, 100 Å, Thermo Scientific) typically running a gradient from 98% solvent A (water, 0.1% formic acid) and 2% solvent B (80% acetonitrile, 20% water, 0.1% formic acid) to 42% solvent B within 95 min and to 65% solvent B within the next 26 min at a flow rate of 300 nl/min (solvents and acids from Fisher Chemicals).

Nano ESI mass spectrometry—Orbitrap Velos Pro (Thermo Scientific): Chromatographically eluting peptides were on-line ionized by nano-electrospray (nESI) using the Nanospray Flex Ion Source (Thermo Scientific) at 2.4 kV and continuously transferred into the mass spectrometer. Full scans within the mass range of 300–1850 m/z were taken from the Orbitrap-FT analyzer at a resolution of 30.000 with parallel data-dependent top 15 MS2-fragmentation with the LTQ Velos Pro linear ion trap (CID). LCMS method programming and data acquisition were performed with the software XCalibur 2.2 (Thermo Fisher). The precursor mass tolerance was 10 ppm while the fragment tolerance was 0.6 Da. The experiments had full trypsin enzyme specificity with 2 as a maximum missed cleavage sites, the FDR target value was 0.01 PSMs.

MS/MS2 data processing for protein analysis and identification was done with the Proteome Discoverer 2.2 (PD, Thermo Scientific) software using the SequestHT and Mascot peptide analysis algorithm(s) and organism-specific taxon-defined protein databases extended by the most common contaminants. We have used the AspGD (Aspergillus Genome Database) to obtain functional annotations, which consist of 29873 entries for the *A. nidulans* FGSC A4 genome. STY phosphorylation was considered as a variable modification, and phospoRS was used to calculate site probabilities with a cut off value of 0.8 [60].

Experiments were performed in triplicates for each time point and proteins with at least two unique peptides identified in each replicate were further considered. Proteins identified in the GFP-only control (AGB551 as genetic background strain over-expressing free GFP controlled by *GPDH* promoter) were discarded for further considerations as putative interaction partners.

## Minimal inhibitory concentration (MIC)

The MIC of the GSK-3β inhibitor VII was determined by growing strains from $10^4$ conidia for 48 h in 96-well plates containing 200 μL glucose MM/well and increasing concentrations of the GSK (glycogen synthase kinase) inhibitor. Next, optical density (O.D.) was read at 600 nm and the percentage of growth calculated with reference to the control, drug-free condition (considered 100% growth).

## Statistical analysis

All statistical analyses were carried out in Prism GraphPad. Statistical analysis was performed for all three biological replicates using a two-way ANOVA multiple comparisons test or a one-tail t-test with statistical significance of $^*$p<0.05, $^{**}$p<0.01 and $^{***}$p<0.001, comparing everything to the wild-type strain in the same condition.

## Supporting information

**S1 Fig. MAPK deletion strains are sensitive to the oxidative stress-inducing compound acrolein.** Heat map and values depicting average radial growth of three biological replicates of protein kinase deletion strains that were significantly sensitive or resistant to at least one concentration of acrolein. Strains were grown from $10^5$ spores for 5 days at 37˚C before radial diameter was measured. The results are expressed as percentage of growth in the presence of acrolein when compared to the drug-free, control medium (defined as 100% growth for each strain). Statistical analysis was performed using a one-tailed, paired t-test when compared to the control condition ($^*$, $p < 0.01$ and $^{**}$, $p < 0.001$).
(TIF)

**S2 Fig. SakA cellular localization is carbon source-dependent.** Microscopy pictures of SakA-GFP hyphae, taken after 16 h growth at 22˚C in xylan minimal medium (MM) and after the addition of glucose for 30 min, show localization in the nucleus. Pictures were taken at different wavelengths (DIC = differential interference contrast, GFP = green fluorescent protein, Hoechst = Hoechst 33258 nucleic acid stain and merged) and scale bars are indicated (a). Percentage of SakA-GFP nuclear localization in different conditions. SakA-GFP was grown as specified in (a) before nuclei with and without GFP were counted for 100 hyphal germlings for each condition and the % of SakA-GFP localization was calculated. Hyphae were stained with Hoechst 33258 in order to confirm GFP nuclear localization (b).
(TIF)

**S3 Fig. GFP-tagged strains, that were constructed for mass spectrometry and phosphorylation site identification, are functional.** The Δste7, ΔmpkB and ΔpbsA strains were complemented with *ste7-gfp*, *mpkB-gfp* and *pbsA-gfp* respectively, and the same GFP-constructions were transformed into the wild-type (WT) background strain. Strains were grown from $10^5$ spores on xylose or glucose minimal medium (MM) supplemented with increasing concentrations of 2-deoxy-glucose (2DG) and allyl alcohol (AA), respectively, for 5 days at 37˚C before pictures were taken (a). The presence of full length GFP-tagged proteins was confirmed by Western blot after 24 h growth in xylan (carbon catabolite de-repressing condition) MM and after the addition of glucose (carbon catabolite repressing condition) for 30 min. Protein levels were normalized by β-actin (GFP/β-actin ratios are indicated) (b).
(TIF)

**S4 Fig. Summary of phenotypes for SakA phosphorylation, PKA and xylanase activities observed for the mitogen-activated protein kinases (MAPKs).** Heat map depicting SakA

phosphorylation levels (P-p38/actin ratio), PKA activity (proportional arbitrary units), xylanase activity (mU x mL x mg dry weight) in the WT, *Δste7*, *ΔmpkB*, *ΔpbsA*, *IgskA* and PbsA$^{S22A}$ $^{S179A}$ strains in the presence of carbon catabolite (CC)-de-repressing and CC-repressing conditions. Numbers represent the average of the results from at least three biological replicates and the heat map color scale is also indicated. Also shown are representative images of the radial growth for each strain in the presence of glucose minimal medium.
(TIF)

**S1 Table. Identification of Ste7-GFP, MpkB-GFP and PbsA-GFP interaction partners by mass spectrometry (MS).** List of proteins identified by MS as interaction partners for Ste7-GFP, MpkB-GFP and PbsA-GFP in carbon catabolite de-repressing (xylan) and repressing (xylan and glucose) conditions. GFP-tagged proteins were immunoprecipitated with GFP-trap beads in the aforementioned conditions, before being submitted to MS analysis.
(XLSX)

**S2 Table. Strains used in this study.**
(DOCX)

## Acknowledgments

We thank the lab members from Georg-University of Gottingen (Germany) for receiving and helping LJA during his 5 months stay in the GHB laboratory and University of São Paulo (Brazil) for additional contributions.

## Author Contributions

**Conceptualization:** Leandro José de Assis, Gustavo Henrique Goldman.

**Data curation:** Leandro José de Assis.

**Formal analysis:** Leandro José de Assis, Lilian Pereira Silva, Kerstin Schmitt, Oliver Valerius, Gerhard H. Braus, Laure Nicolas Annick Ries, Gustavo Henrique Goldman.

**Funding acquisition:** Gerhard H. Braus, Gustavo Henrique Goldman.

**Investigation:** Leandro José de Assis, Lilian Pereira Silva, Li Liu, Kerstin Schmitt, Oliver Valerius, Gerhard H. Braus, Laure Nicolas Annick Ries.

**Methodology:** Leandro José de Assis, Lilian Pereira Silva.

**Project administration:** Gustavo Henrique Goldman.

**Resources:** Gustavo Henrique Goldman.

**Supervision:** Gerhard H. Braus, Laure Nicolas Annick Ries, Gustavo Henrique Goldman.

**Validation:** Leandro José de Assis.

**Visualization:** Leandro José de Assis, Laure Nicolas Annick Ries.

**Writing – original draft:** Leandro José de Assis, Laure Nicolas Annick Ries, Gustavo Henrique Goldman.

**Writing – review & editing:** Leandro José de Assis, Laure Nicolas Annick Ries, Gustavo Henrique Goldman.

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
