## [Decision Letter · Decision Letter 0]

4 Jun 2020

Dear Dr Goldman,

Thank you very much for submitting your Research Article entitled 'The High Osmolarity Glycerol Mitogen-Activated Protein Kinase regulates glucose catabolite repression in filamentous fungi' to PLOS Genetics. Your manuscript was fully evaluated at the editorial level and by independent peer reviewers. The reviewers appreciated the attention to an important problem, but raised some substantial concerns about the current manuscript. Based on the reviews, we will not be able to accept this version of the manuscript, but we would be willing to review again a much-revised version. We cannot, of course, promise publication at that time.

Should you decide to revise the manuscript for further consideration here, your revisions should address the specific points made by each reviewer. We will also require a detailed list of your responses to the review comments and a description of the changes you have made in the manuscript. In particular, I think some of reviewer 1 and 3's comments and request of additional experimental data (e.g., missing controls, potential problems in kinase and glucose uptake assay, verifying CreA-GFP strains, and more mechanistic data showing the interplay between CreA and SakA, etc) are very critical for making more convincing conclusions from this study. Furthermore, we strongly recommend that the revised manuscript be very carefully proofread, because the reviewers and editors found several typos and grammatical errors in the manuscript.  

If you decide to revise the manuscript for further consideration at PLOS Genetics, please aim to resubmit within the next 60 days, unless it will take extra time to address the concerns of the reviewers, in which case we would appreciate an expected resubmission date by email to plosgenetics@plos.org.

[LINK]

We are sorry that we cannot be more positive about your manuscript at this stage. Please do not hesitate to contact us if you have any concerns or questions.

Yours sincerely,

Yong-Sun Bahn, Ph.D.

Guest Editor

PLOS Genetics

Gregory P. Copenhaver

Editor-in-Chief

PLOS Genetics

Reviewer's Responses to Questions

**Comments to the Authors:**

Reviewer #1: Carbon catabolite repression CCR, is an important regulatory process in fungi and bacteria and is of particular important in fungal biology. Here the authors utilized a protein kinase deletion collection in the fungus Aspergillus nidulans to identify protein kinases involved in CCR. The work complements existing observations linking protein kinase activity and regulation of CCR in many organisms. The manuscript contains a significant amount of data that concludes the HOG pathway regulates CCR in fungi. While the conclusions are reasonable based on the data presented, some experiments are missing key controls and leave alternative conclusions in play. Some methods used are notorious for giving spurious results (e.g. kinase assay kits). Also - difference between 2-DG and AA results seem to not be considered as both impact stress responses in the cell beyond CCR that likely are mediated by HOG and related kinases. A lot of emphasis is placed on single assays such as CreA-GFP localization and phosphorylation of SakA. Additional evidence of SakA activation and CreA activity are needed to improve the rigor of the studies.

-Figure 1 data is important for the study – but some consideration discussion of 2-DG off-target effects and AA generation of oxidative stress should be considered. Are AA R/S strains also R/S to oxidative stress?

-The assay to measure glucose uptake is not a true glucose uptake assay. The author’s measured glucose in the culture media after incubation with the fungus. The changes in glucose levels in the media may be due to differences in uptake, but also maybe not. To measure uptake, radio-labeled glucose is needed. Since the mutants have growth defects, it is unclear if the differences in the media are simply due to the reduced growth of the mutants and lack of glucose utilization.

-Fig. 2A – unclear what the STRING analyses is presenting. Are these direct physical interactions? Indirect genetic interactions? More details on this analysis are needed.

-For the western blots of SakA phosphorylation – total SakA is a critical control that is missing. Without that control, interpretation of these data are tenuous. Also, a sakA null mutant is an important control here.

-Lines 204/205 – Xylose is listed as being both de-repressing and repressing

-Is phosphorylation of SakA sufficient evidence to conclude that activation is happening or inhibited in these conditions? Does SakA localization change in repressing and de-repressing conditions?

-Kinase activity assays such as those in Figure 2C need mutant controls – these assays are notorious for non-specific kinase activity. What is the PKA activity in a PKA mutant strain?

-Figure 3A – these are not the best sub-cellular localization images. Are they representative? The nuclei are not clear and no co-localization is shown.

-Figure 4A – approximate size of bands needed; loading control is hard to read, why not use the actin antibody here? There seems to be a lot of variability in protein loaded.

-Figure 5 – these are important data, but they are preliminary without confirmation via rigorous Co-IP experiments and for the phosphoproteomics, confirmation that the phosphosites are indeed functional.

-has this gskA inhibitor been validated in fungi? Perhaps an inducible gskA strain is a more rigorous approach to support the conclusions.

-How do changes in carbon sources used in the study impact the osmolarity of the system? Is this activation of HOG really reflecting changes in osmolarity?

Reviewer #2: The authors started off with a mutant collection of non-essential protein kinases in Aspergillus nidulans because the activity of the main regulator of carbon catabolite repression, CreA, is regulated by phosphorylation. Studying potential roles in carbon catabolite repression they discovered that the central regulator CreA is able to form protein complexes with several other regulators. Apparently, the protein complex is a scavenger for free CreA, which upon dissociation can fulfill its function. This work nicely demonstrates the cross-talk between carbon utilization and different signaling pathways in the cell.

This is one of the few papers where I don’t have any major suggestions for improvement. Excellent paper. However, I am not sure whether the proposed large protein complex as displayed in Fig. 8 is not an oversimplification. I don’t think the data really show that such a large protein complex exists. The observed interactions could be transient interactions of different protein combinations. Or did I oversee something?

Reviewer #3: PGENETICS-D-20-00734

In fungal organisms, carbon catabolite repression (CCR) is a regulatory mechanism that inhibits enzymes for the consumption of secondary carbon sources, when a preferred primary carbon source such as glucose is present. CCR has been studied for long in diverse fungal genetic model systems, including Aspergillus nidulans. This for example led to the discovery of transcription factor (TF) CreA, which is highly conserved in fungal organisms. The regulation of CreA is mediated by phosphorylation at different residues. This complexity of the regulation of carbon source utilisations is only partly understood, including the interplay between different kinase signaling pathways.

In this paper, different mitogen-activated protein kinases (MAPKs) were identified that are crucial for CCR and protein kinase A (PKA) activity, which is essential for carbon source utilisation. With this, the manuscript contributes substantially to our understanding of the crosstalk between different signaling pathways.

The paper starts with the screening of a library, containing non-essential protein kinase (NPK) deletion strains, to identify kinases important for CCR. This resulted in the identification of several MAPK pathway kinases. This is followed by construction of a network interaction profile of 23 protein kinases, and a microscopic investigation of MAPKs dependent nuclear CreA localization. Finally, MS and phosphoproteome analysis identifies interaction partner of MAPKS and the effect of MAPK phosphorylation on osmotic stress resistance, HOG pathway activation and enzyme activities.

At the end of the ms, the authors contribute a model, explaining the correct integration of signaling events and cascades that constitute the different pathways involved in glucose-mediated CCR as well as CC-de-repression.

The data presented are of high quality and the ms addresses the major question how different carbon sources are utilized on the molecular level. As such, the ms deserves publication.

However, I would like to suggest several improvements:

1. Important information is missing in the Methods section: the construction of phosphorylation mutants (line 388) should be included. In case of Data Processing (line 633 - 638), I recommend to give the following information in the Methods section:

- Database size and origin (e.g. Uniprot, Refseq?)

- Precursor and fragment mass tolerance

- Enzyme specificity

- Which FDR target value was used and to which level was it applied (PSMs, Peptides, Proteins)?

- Was a phosphoRS cutoff value applied?

2. line 249ff: The experiments for Fig. 4a should be described in more detail. For example, carry all strains single copies of the gfp-creA construct? Further, how were the CreA-GFP protein levels measured? Did the authors take also in account the CreA-GFP degradation products, or did they consider only the full size CreA-GFP protein? One may conclude from the protein gel that single lanes contain different protein amounts.

3. I miss mentioning PkaC and PkaR, both which are present in the model of Figure 8, but are not described in the ms.

4. I suggest to give a list of abbreviations of the many genes/proteins for better readability of the ms

5. Some formal points:

- Fig. 3a: It should read “Hoechst”, not Hoescht

- Fig. 3b: Why do they show a 125% y-axis? More than 100 % should be impossible. It should read “Microscopy”

- Fig. 5d: “Prediction”, instead of “Predition”

- Check the list of references, which is not consistently abbreviated, e.g. Fungal Gen Biol or Fungal Genet Biol or Fungal Genet Biol. Elsevier Inc.

**Have all data underlying the figures and results presented in the manuscript been provided?**

Reviewer #1: Yes

Reviewer #2: Yes

Reviewer #3: Yes

PLOS authors have the option to publish the peer review history of their article (what does this mean?). If published, this will include your full peer review and any attached files.

Reviewer #1: No

Reviewer #2: No

Reviewer #3: No

---

## [Decision Letter · Decision Letter 1]

15 Jul 2020

Dear Dr Goldman,

We are pleased to inform you that your manuscript entitled "The High Osmolarity Glycerol Mitogen-Activated Protein Kinase regulates glucose catabolite repression in filamentous fungi" has been editorially accepted for publication in PLOS Genetics. Congratulations!

Yours sincerely,

Yong-Sun Bahn, Ph.D.

Guest Editor

PLOS Genetics

Gregory P. Copenhaver

Editor-in-Chief

PLOS Genetics

Comments from the reviewers (if applicable):

Two referees, who made critical comments for the original manuscript, re-evaluated the revised version of the manuscript. Both referees highly appreciated the newly added data, including localization of SakA and inclusion of proper controls, and agreed that the authors properly addressed the previous comments.

Reviewer's Responses to Questions

**Comments to the Authors:**

Reviewer #1: The author's have responded positively the previous critiques. The localization of sakA in response to carbon source is nice further evidence of a role for this pathway in CCR. Additional controls have been added to key experiments which strengthens conclusions. Congratulations to the authors on a nice study.

Reviewer #3: I have carefully checked all responses to my previous review. The authors have done the corrections in the revised manuscript as requested.

**Have all data underlying the figures and results presented in the manuscript been provided?**

Reviewer #1: Yes

Reviewer #3: Yes

PLOS authors have the option to publish the peer review history of their article (what does this mean?). If published, this will include your full peer review and any attached files.

Reviewer #1: No

Reviewer #3: No

**Data Deposition**

http://datadryad.org/submit?journalID=pgenetics&manu=PGENETICS-D-20-00734R1

**Press Queries**

---

## [Editor Report · Acceptance letter]

19 Aug 2020

PGENETICS-D-20-00734R1 

The High Osmolarity Glycerol Mitogen-Activated Protein Kinase regulates glucose catabolite repression in filamentous fungi 

Dear Dr Goldman, 

We are pleased to inform you that your manuscript entitled "The High Osmolarity Glycerol Mitogen-Activated Protein Kinase regulates glucose catabolite repression in filamentous fungi" has been formally accepted for publication in PLOS Genetics! Your manuscript is now with our production department and you will be notified of the publication date in due course.

With kind regards,

Jason Norris

PLOS Genetics

On behalf of:
